# High performance TadA-8e derived cytosine and dual base editors with undetectable off-target effects in plants

Tingting Fan [1,2,7], Yanhao Cheng[3,7], Yuechao Wu [4,5,7], Shishi Liu[1,7], Xu Tang[2,7], Yao He[1], Shanyue Liao[1], Xuelian Zheng [2], Tao Zhang [4,5] ✉, Yiping Qi [3,6] ✉ & Yong Zhang [2] ✉

Cytosine base editors (CBEs) and adenine base editors (ABEs) enable precise C-to-T and A-to-G edits. Recently, ABE8e, derived from TadA-8e, enhances A-to-G edits in mammalian cells and plants. Interestingly, TadA-8e can also be evolved to confer C-to-T editing. This study compares engineered CBEs derived from TadA-8e in rice and tomato cells, identifying TadCBEa, TadCBEd, and TadC-BEd_V106W as efficient CBEs with high purity and a narrow editing window. A dual base editor, TadDE, promotes simultaneous C-to-T and A-to-G editing. Multiplexed base editing with TadCBEa and TadDE is demonstrated in transgenic rice, with no off-target effects detected by whole genome and transcriptome sequencing, indicating high specificity. Finally, two crop engineering applications using TadDE are shown: introducing herbicide resistance alleles in *OsALS* and creating synonymous mutations in *OsSPL14* to resist *OsMIR156*-mediated degradation. Together, this study presents TadA-8e derived CBEs and a dual base editor as valuable additions to the plant editing toolbox.

Cytosine base editors (CBEs) represent a revolutionary gene-editing tool based on the nCas9 protein and a cytidine deaminase, enabling precise C-to-T base editing[1]. Prior research has successfully harnessed various cytidine deaminases from animals and humans, including rAPOBEC1[2–5], hAID[6], PmCDA1[5,7,8], hAPOBEC3A[9,10], and hAPOBEC3B[11], to achieve targeted C-to-T base editing. The efficiency and product purity of C-to-T base editing have been further improved by incorporating one or multiple copies of uracil DNA glycosylase inhibitor (UGI)[2,12,13]. CBEs offer versatility in introducing non-sense mutations for genetic knockouts[14–16], conferring gain-of-function traits such as herbicide resistance[17,18], and fine-tuning gene expression by modulating cis-elements[19]. Recent studies have reported potential genome-wide off-target effects when utilizing BE3 based on rAPOBEC1 in mouse embryos[20] and rice[21]. Engineered Cytosine Base Editors, including eA3A-BE3[22], A3G-BEs[23], YE1-BE3-FNLS[24], hA3B[11], and PmCDA1[25], were demonstrated to mitigate off-target effects in humans and plants. Adenine Base Editors (ABEs) are based on nCas9 and an evolved tRNA adenosine deaminase, enabling precise A-to-G base conversions[26]. At

[1]Department of Biotechnology, School of Life Sciences and Technology, Center for Informational Biology, University of Electronic Science and Technology of China, Chengdu 610054, China. [2]Chongqing Key Laboratory of Plant Resource Conservation and Germplasm Innovation, Integrative Science Center of Germplasm Creation in Western China (Chongqing) Science City, School of Life Sciences, Southwest University, Chongqing 400715, China. [3]Department of Plant Science and Landscape Architecture, University of Maryland, College Park, ML 20742, USA. [4]Jiangsu Key Laboratory of Crop Genomics and Molecular Breeding/Zhongshan Biological Breeding Laboratory/Key Laboratory of Plant Functional Genomics of the Ministry of Education, College of Agriculture, Yangzhou University, Yangzhou 225009, China. [5]Jiangsu Co-Innovation Center for Modern Production Technology of Grain Crops/Jiangsu Key Laboratory of Crop Genetics and Physiology, Yangzhou University, Yangzhou 225009, China. [6]Institute for Bioscience and Biotechnology Research, University of Maryland, Rockville, ML 20850, USA. [7]These authors contributed equally: Tingting Fan, Yanhao Cheng, Yuechao Wu, Shishi Liu, and Xu Tang. ✉e-mail: zhangtao@yzu.edu.cn; Yiping@umd.edu; zhangyong916@swu.edu.cn

present, the main variants of TadA in ABEs include TadA-7.10, TadA-8e, and TadA-9, exhibiting highly efficient A-to-G base editing in human cells[27–29] and plants[30,31]. Recent research has revealed genome-wide off target effects of TadA-8e in the rice genome[8,25] and of TadA9 in the rice transcriptome[32]. These findings highlight the importance of future research to improve editing specificity of these highly active ABEs. Dual base editors recruit both cytidine deaminase and adenosine deaminase via the CRISPR-nCas9 system, enabling simultaneous C-to-T and A-to-G base editing[1,33]. Dual base editors are powerful tools for targeted saturation mutagenesis and directed evolution in crops[34]. Currently, there is limited exploration of dual base editors in plants, with notable systems including STEME[35], SIWSS[36], and MoBE[37]. These systems require the use of both deaminases simultaneously, resulting in larger molecular weights of the base editors which will hinder ribonucleoprotein (RNP) delivery for transgene-free genome editing in plants[38]. Furthermore, the different editing windows of the two distinct deaminases compromise editing precision. These current limitations can be addressed by utilizing a single deaminase to achieve simultaneous C-to-T and A-to-G base editing, providing a potential improvement in precision, and overcoming the challenges associated with the existing systems. Recently, a few studies reported a series of base editors obtained via the further engineering of TadA-8e, including TadCBEa, TadCBEd, TadCBEd_V106W, TadDE, eTd-CBE, and Td-CBEmax[39–41]. These base editors confer C-to-T base editing or simultaneous C-to-T and A-to-G editing in human cells. In this study, we investigate whether these CBEs and dual base editors could also function in plants. We conduct whole genome sequencing and transcriptome sequencing of the edited plants to elucidate the editing efficiency and specificity. In addition, we showcase two powerful applications of the emerging dual base editor, TadDE, for engineering gain-of-function traits in plants.

## Results

### Comparison of multiple base editors in rice cells via multiplexed editing

The cytosine base editor consists of nCas9 and cytidine deaminase, while the adenine base editor comprises nCas9 and adenosine deaminase[34]. Fusion of nCas9, cytidine deaminase, and adenosine deaminase allows simultaneous C-to-T and A-to-G editing[35–37]. However, these dual-base editors have larger molecular weight, lower editing efficiency, and significant differences in editing windows due to the characteristics of deaminases. Researches have shown that engineered TadA-8e-derived cytosine base editors (TadCBEs) and dual-base editors (TadDE) can achieve C-to-T and C-to-T&A-to-G editing, respectively, in human cells[39,40]. In order to investigate whether the TadA-8e derived CBEs (TadCBEs) can effectively perform C-to-T mutations in plants, we constructed a series of promising TadCBEs for plant expression, including TadCBEa, TadCBEd, TadCBEd_V106W, eTd-CBE, and Td-CBEmax[39,40]. Also, the dual base editor, TadDE[40], was constructed with nCas9 and nCas9-NG, to target NGG and NG protospacer adjacent motifs (PAMs), respectively. A3A_Y130F (as an efficient CBE) and ABE8e (as an efficient ABE) were included as controls. These nine base editing systems (Fig. 1a and Supplementary Fig. 1 and Supplementary Table 1) were tested for editing of 20 endogenous sites in the rice genome. For each base editor, two multiplexed T-DNA vectors were constructed to each target 10 target sites in the rice genome (Supplementary Table 2). We conducted transient transformation of rice protoplasts to assess the editing performance of the base editing systems via next generation sequencing (NGS) of PCR amplicons.

Through the analysis of NGS data, we found that most TadCBEs and TadDE could achieve effective C-to-T base editing in rice cells (Fig. 1b). Upon comprehensive analysis of all 20 target sites, the C-to-T base editing efficiencies of TadCBEa, TadCBEd, and TadCBEd_V106W were 4.5% ~ 55.4%, 8.0% ~ 47.2%, and 5.0% ~ 53.7%, respectively, which were comparable to that of A3A_Y130F (9.0% ~ 53.0%) (Fig. 1c).

In comparison, Td-CBEmax showed slightly lower C-to-T base editing efficiency while eTd-CBE showed the lowest C-to-T editing efficiency among the CBEs (Fig. 1c). TadCBEa and TadCBEd showed detectable residual A-to-G editing in some target sites (Fig. 1b, d). Furthermore, the insertion and deletion (indel) efficiencies by all these base editors were very low, within the range of sequencing error rates (Fig. 1e). Taken together, these data suggest TadA-8e derived CBEs are promising CBEs in achieving high cytosine base editing efficiency and purify.

Among the 20 target sites, simultaneous C-to-T and A-to-G editing was detected at 16 sites by TadDE. Analysis of the sequences of the remaining four sites which did not show A-to-G editing revealed that there were no editable A within the editing window (Fig. 1b and Supplementary Fig. 2 and 3). Remarkably, TadDE showed A-to-G base editing efficiencies (0.1% ~ 40.8%) that were comparable to those of ABE8e (0.1% ~ 24.5%) (Fig. 1d). Hence, TadDE indeed conferred highly efficient simultaneous C-to-T and A-to-G base editing. Since TadDE-NG prefers NG PAMs, it is not surprising to see compromised editing activity at the NGG PAM sites (Fig. 1c, d), consistent with our previous observation on Cas9-NG's nuclease activity in rice cells[42]. However, TadDE-NG would allow simultaneous base editing at relaxed 5'-NG-3' PAM sites.

To further investigate base editing by TadDE, we grouped the target sites based on A-to-G editing efficiencies, using 2% as the break point. This analysis showed that target sites with higher A-to-G editing efficiencies by TadDE also showed higher A-to-G editing efficiencies by ABE8e (Supplementary Fig. 4a), and the opposite was also true (Supplementary Fig. 4b). These data suggest that low editing efficiencies observed at some sites by TadDE were due to poor protospacer choices.

### Base editing profiles of TadA-8e derived CBEs and dual base editor in rice cells

Using the NGS data, we further analyzed the genome editing profiles of the 20 target sites and presented the data in heat maps (Supplementary Fig. 5 and 6) and base percentages (Supplementary Fig. 7) at editable cytosine and adenine nucleotides. In comparison to the C-to-T editing window spanning positions 3rd to 16th of A3A_Y130F, the C-to-T editing windows of TadCBEa, TadCBEd, and TadCBEd_V106W are narrower, concentrated at positions 4th to 8th of the protospacers, which is the A-to-G base editing window of ABE8e (Fig. 2a). Furthermore, the C-to-T base editing window and A-to-G base editing window by TadDE nearly overlapped (Fig. 2a). These data suggest that the engineered TadA-8e deaminases, despite altered base editing preference, have retained the general base editing window of TadA-8e.

Detailed analysis of NGS reads also revealed top base editing genotypes by each base editor in rice protoplasts. For the three TadA-8e derived CBEs, the most frequent reads at the 20 target sites all contained C-to-T base edits (Supplementary Figs. 8–10). However, the top edits at the *OsCDC48*-sgRNA01 and *OsDEP1*-sgRNA01 sites by TadCBEa were simultaneous C-to-T and A-to-G base editing (Supplementary Fig. 8). For TadCBEd, only one top edit at the *OsEV*-sgRNA01 site was simultaneous C-to-T and A-to-G editing, and the remaining top edits were all pure C-to-T editing (Supplementary Fig. 9). For TadCBEd_V106W, all top edits were pure C-to-T editing (Supplementary Fig. 10). These data indicate an overall high C-to-T base editing purity of these TadA-8e derived CBEs. Observational data suggest a potential positive correlation between the capability for residual A-to-G editing and the functional performance of the adenine deaminase originating from TadA-8e.

The dual base editor TadDE generated simultaneous C-to-T and A-to-G editing reads at 18 out of the 20 target sites (Fig. 2b and Supplementary Fig. 11). At eight target sites, the top edits were simultaneous C-to-T and A-to-G edits (Fig. 2b). Hence, TadDE is indeed a potent dual base editor. Interestingly, we observed C-to-G

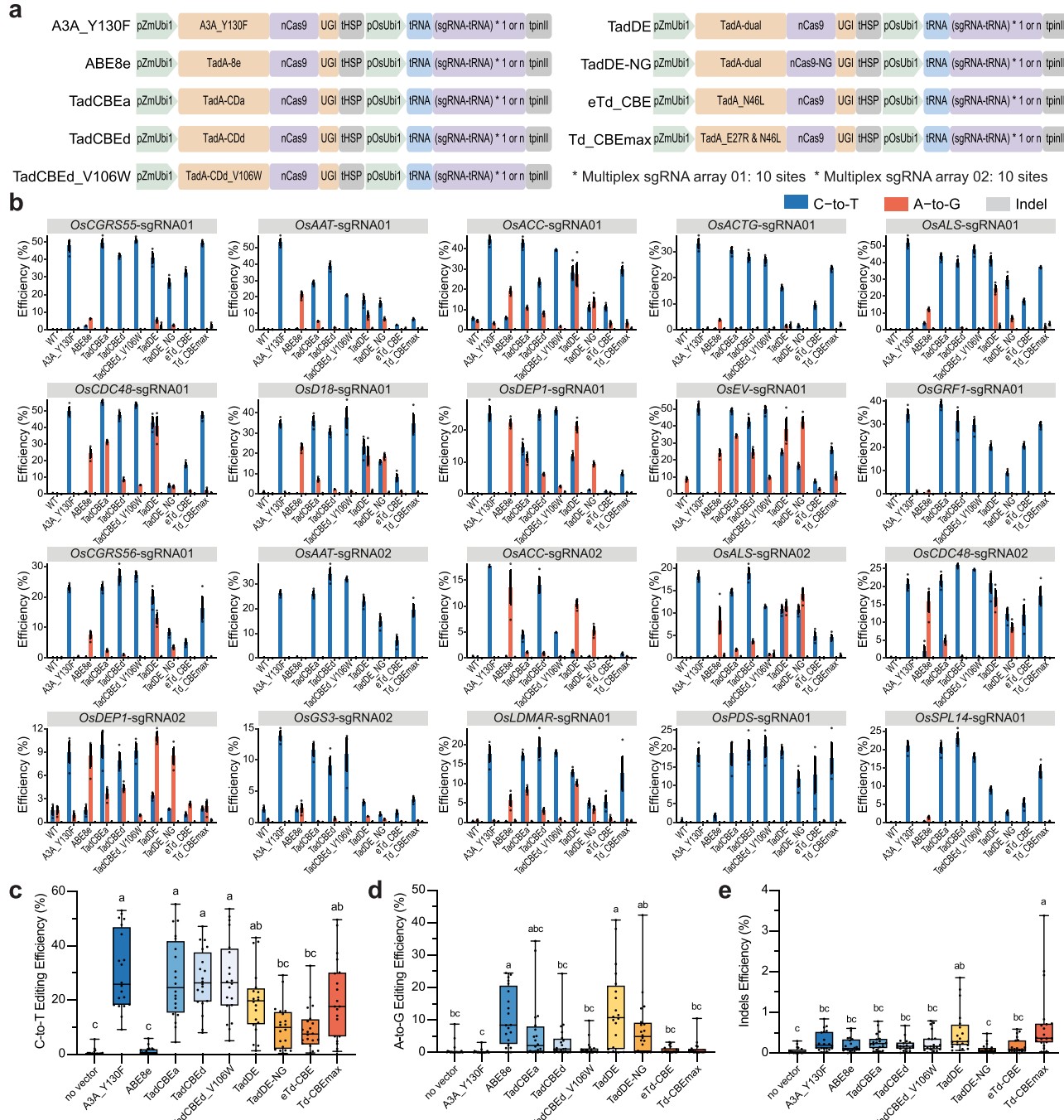

**Fig. 1 | Comparison of TadA-8e derived base editors for C-to-T, A-to-G and dual base editing in rice cells. a** Major base editors evaluated in this study along with their corresponding schematic diagrams. **b** Characterization of different base editors in rice protoplasts. C-to-T base editing is shown in blue, A-to-G base editing is shown in orange, and indel mutation is shown in gray. Dots represent individual values, and bars represent mean ± SD of three biological replicates. **c** C-to-T editing efficiency of tested base editors in rice protoplasts at 20 target sites. Each dot represents the average of three biological replicates. Different letters indicate significant differences (*P* < 0.05; one-way ANOVA, Duncan test). The maxima, centre, and minima of box the refer to Upper quartile, median, and Lower quartile. The maxima and minima of whiskers refer to maximum value and minimum value.

**d** A-to-G editing efficiency of tested base editors in rice protoplasts at 20 target sites. Each dot represents the average of three biological replicates. Different letters indicate significant differences (*P* < 0.05; one-way ANOVA, Duncan test). The maxima, centre, and minima of box the refer to Upper quartile, median, and Lower quartile. The maxima and minima of whiskers refer to maximum value and minimum value. **e** Indels efficiency of tested base editors in rice protoplasts at 20 target sites. Each dot represents the average of three biological replicates. Different letters indicate significant differences (*P* < 0.05; one-way ANOVA, Duncan test). The maxima, centre, and minima of box the refer to Upper quartile, median, and Lower quartile. The maxima and minima of whiskers refer to maximum value and minimum value. Source data are provided as a Source Data file.

and simultaneous C-to-T and C-to-G base editing at the *OsALS*-sgRNA01 site by all four TadA-8e derived base editors (Supplementary Figs. 8–11), suggesting these base editors can induce C-to-G base editing in certain sequence context.

## Evaluation of six base editors in tomato cells via multiplexed editing

Having assessed the emerging TadA-8e derived base editors in rice cells, we generated four Gateway entry clones for TadCBEa, TadCBEd,

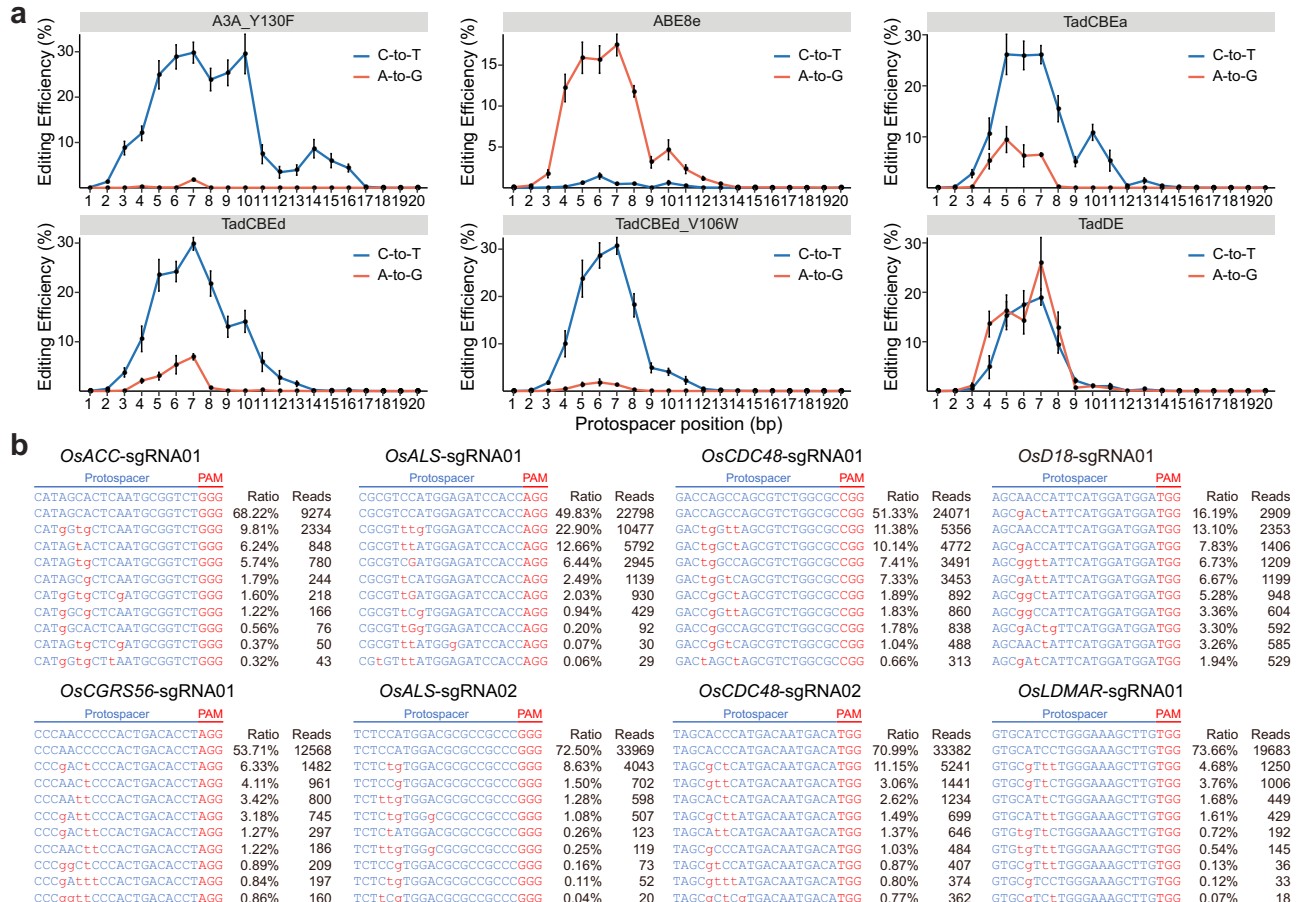

**Fig. 2 | Base editing profiles by different base editors in rice cells. a** Base editing activity window for A3A_Y130F-CBE, ABE8e, TadCBEa, TadCBEd, TadCBEd_V106W and TadDE across twenty different target sites in rice cells. The editing efficiencies of C-to-T and A-to-G base editing at 20 sites, ranging from positions 1–20, were fitted together to obtain the editing window as shown in the figure. Dots represent average editing across all sites containing the specified bases at the indicated position within the protospacer. Data are presented as mean values ± SEM. Individual data points used for this analysis are included in Supplementary Fig. 2 and 3. **b** Genotyping of protoplasts shows TadDE promotes simultaneous C-to-T and A-to-G base editing in rice cells. Sequences in red indicate base editing outcomes. The values in right represent ratio and reads of mutation alleles. See Supplementary Fig. 11 for results at 12 additional target sites. Source data are provided as a Source Data file.

TadCBEd_V106W, and TadDE (Supplementary Fig. 12a), which would enable their applications in other plant species. To evaluate the effectiveness of these CBEs and dual base editor in dicots, we tested them in tomato protoplasts. A3A_Y130F and ABE8e were also included as controls for C-to-T and A-to-G editing, respectively. Six multiplexed base editing T-DNA vectors were constructed for transformation of tomato protoplasts, followed by NGS analysis of PCR amplicons to assess editing efficiencies at four target sites. Our results indicated that TadCBEs and TadDE were effective in achieving C-to-T editing in tomato cells across four target sites (Fig. 3a). The editing efficiencies for TadCBEa, TadCBEd, TadCBEd_V106W, and TadDE were 8.9 ~ 23.2%, 9.8 ~ 26.4%, 7.5 ~ 20.2%, and 6.9 ~ 24.9%, respectively, slightly lower than that of A3A_Y130F (17.2% to 31.4%) (Fig. 3a), but the difference was not statistically significant (Fig. 3b).

Regarding A-to-G editing, all tested base editors exhibited low editing efficiencies at sites *SolyA7*-sgRNA03 and *SolyA7*-sgRNA04 (Fig. 3a). A closer examination of the target sequences at these sites revealed limited editable adenines within the editing window. Based on the NGS data, TadCBEs induced low A-to-G editing efficiencies between 0.7% and 4.2%. TadDE showed effective A-to-G editing (1.3% to 10.0%), albeit lower than ABE8e (1.1% to 22%) (Fig. 3c). Based on the overall analysis of C-to-T editing (Fig. 3b) and A-to-G editing (Fig. 3c), TadDE was also benchmarked as a dual base editor in tomato.

Interestingly, we detected elevated indel frequency by A3A_Y130F (Fig. 3d), which could be partly due to its slightly higher deaminase activity (Fig. 3a, b)[13]. Impressively, indel frequencies by all TadA-8e derived base editors were very low, indistinguishable from the baseline sequencing error rate (Fig. 3d).

As with the data in rice protoplasts, A3A_Y130F exhibited a larger base editing window than TadA-8e derived base editors (Fig. 3e). The TadDE showed quite overlapping C-to-T and A-to-G base editing windows (Fig. 3e). Indeed, simultaneous base editing was observed in the top base edits at *SlBlc*-sgRNA01 and *SlBlc*-sgRNA02 target sites (Fig. 3f), where high efficiency A-to-G editing by ABE8e was also observed (Fig. 3a, c and Supplementary Fig. 12b).

### Singular and multiplexed base editing in rice plants by TadCBEa and TadDE

We next focused on TadCBEa and TadDE to assess their genome editing outcomes in stable transgenic rice lines, expressing 10 sgRNAs simultaneously (Fig. 4a). Due to the lower emergence of transgenic shoots in the system targeting multiplex array 01 and the resultant decrease in transgenic positivity rate (Supplementary Table 3), T0 lines from multiplex array 02 sites were selected for subsequent analyses of editing efficiency and specificity. Transgenic shoots were initially subjected to Sanger sequencing to confirm base editing at the target

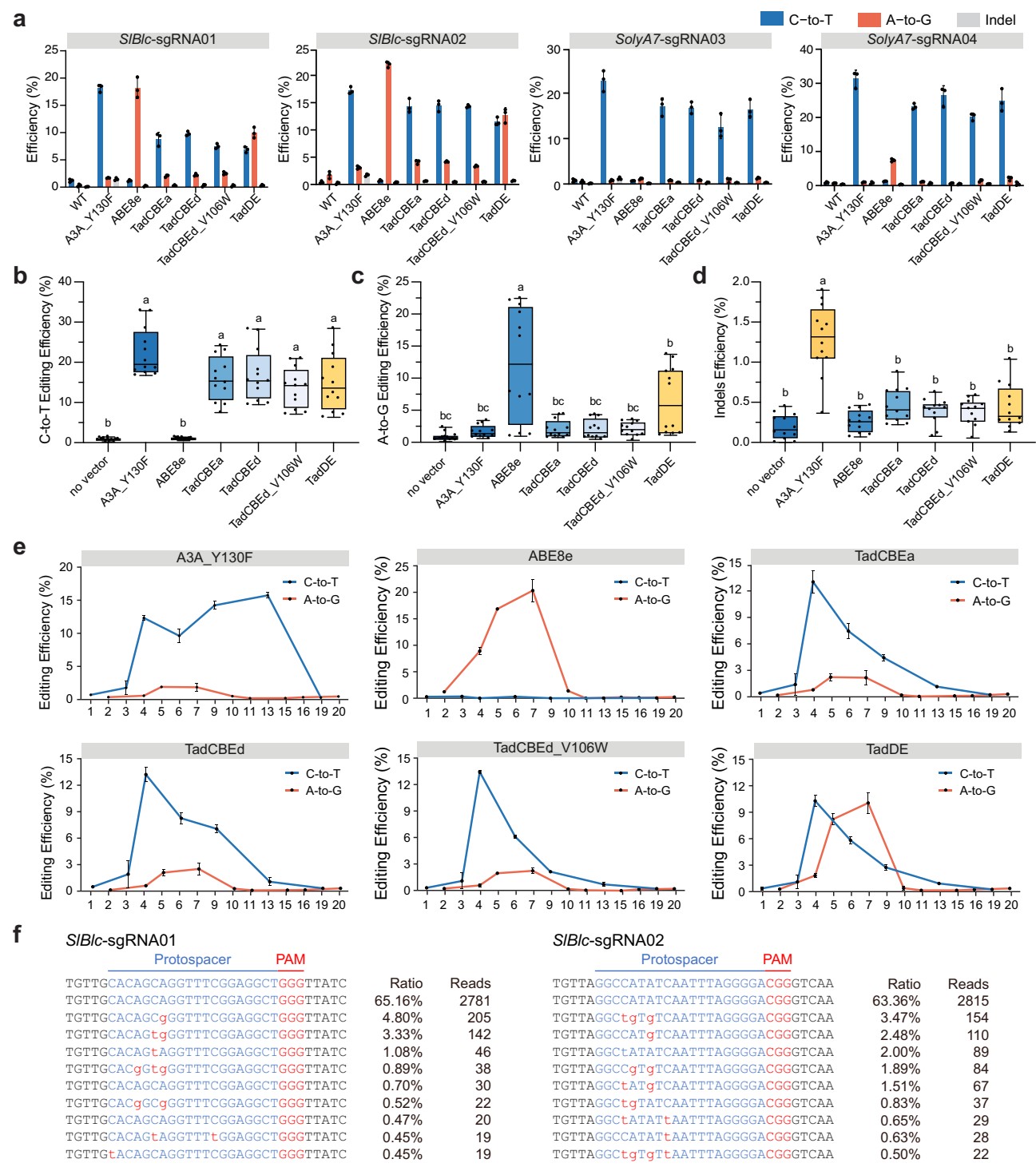

sites (Supplementary Fig. 13), followed by whole genome sequencing (WGS) (Fig. 4b). After aligning WGS data to the reference rice genome, the analysis of editing efficiency in plants revealed that the C-to-T base editing efficiencies of TadCBEa and TadDE in stable plants were relatively comparable (Fig. 4c and Supplementary Table 4), consistent with to the data in protoplasts (Fig. 1b). Notably, high efficiency biallelic C-to-T base editing was found at most target sites by TadCBEa and TadDE (Fig. 4c). Analysis of reads from edited plants confirmed that editing by TadCBEa and TadDE occurred within the editing windows of the target sites (Supplementary Fig. 14 and 15). As expected, low efficiencies of A-to-G base editing were found in plants edited by TadCBEa (Fig. 4d and Supplementary Table 4). Strikingly, barely any A-to-G base editing

could be found in the plants edited by TadDE (Fig. 4d and Supplementary Table 4), which is very surprising. Given that we also observed low transformation efficiency with this TadDE multiplexed editing construct, we hypothesized that the absence of A-to-G editing events in the recovered stable transgenic rice lines might be attributed to potential lethality caused by base editing at one or more genes among the 10 target genes.

To investigate this issue, we performed singular base editing using TadDE at six target sites (Fig. 4e), with four of them included in the multiplexed array 02 editing construct and two in multiplexed array 01. Analysis of transgenic T0 lines showed that simultaneous C-to-T and A-to-G base editing was induced at five out of six target sites,

**Fig. 3 | Assessment of TadA-8e derived cytosine base editors and dual base editor in tomato cells. a** Editing efficiency of TadA-8e derived cytosine and dual base editors at four target sites in tomato. Dots represent individual values, and bars represent mean ± SD of three biological replicates. **b** C-to-T editing efficiency of select base editors in tomato protoplasts at four target sites. Each dot represents a biological replicate. The data for each column include three biological replicates of four tomato loci. Different letters indicate significant differences ($P < 0.05$; one-way ANOVA, Duncan test). The maxima, centre, and minima of box the refer to Upper quartile, median, and Lower quartile. The maxima and minima of whiskers refer to maximum value and minimum value. **c** A-to-G editing efficiency of select base editors in tomato protoplasts at four target sites. Each dot represents a biological replicate. The data for each column include three biological replicates of four tomato loci. Different letters indicate significant differences ($P < 0.05$; one-way ANOVA, Duncan test). The maxima, centre, and minima of box the refer to Upper quartile, median, and Lower quartile. The maxima and minima of whiskers refer to maximum value and minimum value. **d** Indels efficiency of select base editors in tomato protoplasts at four target sites. Each dot represents a biological replicate. The data for each column include three biological replicates of four tomato loci. Different letters indicate significant differences ($P < 0.05$; one-way ANOVA, Duncan test). The maxima, centre, and minima of box the refer to Upper quartile, median, and Lower quartile. The maxima and minima of whiskers refer to maximum value and minimum value. **e** Editing window of six base editors at site *SlBlc*-sgRNA01 and *SlBlc*-sgRNA02. The editing efficiencies of C-to-T and A-to-G base editing at 4 sites, ranging from positions 1–20, were fitted together to obtain the editing window as shown in the figure. Data are presented as mean values ± SEM. **f** Genotyping of protoplasts edited by TadDE at site *SlBlc*-sgRNA01 and *SlBlc*-sgRNA02. Lowercase red letters indicated base editing outcomes. The values on the right represent ratio and reads of mutation alleles. Source data are provided as a Source Data file.

rather efficiently (Fig. 4f, g and Supplementary Fig. 16). However, at the *OsEV*-sgRNA01 site, no C-to-T base editing was found in T0 lines (Fig. 4f), which is in great contrast to the protoplast data, where high frequency simultaneous C-to-T and A-to-G base editing was observed (Fig. 1b and Supplementary Fig. 11). Therefore, we hypothesize that C-to-T editing at the *OsEV*-sgRNA01 site, either individually or in combination with other targeted sites, could induce lethality that prevented the regeneration of genome-edited rice lines with the multiplex array 01. Though, further experiments are needed to fully establish this causal relationship as well as to investigate the causal gene(s) in the multiplex array 02 that also caused lethality upon editing. Nevertheless, our singular base editing experiment demonstrated TadDE's capability of inducing simultaneous base editing in rice stable plants.

## Genome-wide off-target analyses of TadCBEa and TadDE in rice by whole genome and transcriptome sequencing

We also employed WGS to investigate genome-wide off-target effects by TadCBEa and TadDE in rice plants that endured multiplexed editing. With multiplexed editing, we could look for potential gRNA dependent off-target effects of 10 protospacers as well as gRNA-independent off-target effects. In the WGS pipeline, we included tissue culture with Agrobacterium as control types (e.g., transgenic plants) (Fig. 5a). In parallel, a subset of TadCBEa and TadDE edited plants along with the transgenic control plants were used for transcriptome sequencing (Fig. 5a). With RNA-seq, we confirmed the expression of base editors (e.g., Cas9 and the deaminases) in all transgenic plants (Supplementary Fig. 17), and this data support the meaningfulness of using these plants to investigate the potential off-target effects at the genome and transcriptome levels.

Based on our analysis, TadCBEa and TadDE generated similar numbers of indels (~60) and single nucleotide variants (SNVs) (~200) to those of tissue culture control (Fig. 5b, c). The distribution profiles of indels and SNVs in different annotated genomic regions were similar between tissue culture control and edited plants either by TadCBEa or TadDE (Supplementary Fig. 18). The SNVs appeared to be randomly distributed in the rice genome (Supplementary Fig. 19). Further examination of the six SNV types showed comparable trends among the base edited lines and control plants (Fig. 5d). Compared to the tissue culture control, there were not significantly more C-to-T SNVs or A-to-G SNVs in the edited plant population by TadCBEa and TadDE (Fig. 5e, f). Overall, the three sample groups shared similar SNV compositions (Fig. 5g). Further evaluation of nucleotide compositions immediately adjacent to the mutated cytosines and adenines didn't reveal any sequence preference (Fig. 5h), such as the characteristic TA motif that is known to be preferred by TadA-8e[25]. Together these analyses suggest there is lack of significant genome-wide gRNA-independent off-target effects by either TadCBEa or TadDE in rice. The discovered mutations in each by WGS were largely somaclonal variations due to tissue culture, just as we previously reported[13,25,43].

Since we simultaneously targeted 10 endogenous sites for base editing in the rice genome, we wanted to know whether any of the sgRNA had caused gRNA-dependent off-target mutations. Interestingly, allele frequency analysis of discovered indels and SNVs at individual sequenced plants showed that >50% SNVs are likely germline-transmittable (e.g., either heterozygous or homozygous), which was much higher than the percentages of indels (Supplementary Fig. 20). It is important to know whether any of these SNVs were due to off-target effects of our base editors. To this end, we used Cas-OFFinder[44] to screen 20 potential off-target sites with mismatches ≤ 3 among 10 target sites. Then, we tried to match all the C-to-T and A-to-G SNVs discovered in our edited plants to these potential off-target sites. Our analysis revealed that among 20 potential off-target sites, 16 sites showed no detectable editing events in individual plants edited by TadCBEa or TadDE (Supplementary Fig. 21 and Supplementary Table 5). Among the remaining 4 off-target sites, editing events at 3 sites occurred at positions with one mismatch to the target sites (Supplementary Fig. 21 and Supplementary Table 5). Interestingly, such off-target mutations were caused by the same two gRNAs either with TadCBEa or TadDE (Supplementary Fig. 21), suggesting gRNA-dependent off-target mutations by these base editors were simply caused by high sequence similarity between the target site and the off-target sites, a characteristic that we previously benchmarked based on WGS for Cas9 in plants[43].

Since TadA derived deaminases could generate transcriptome-wide off-target mutations when over-expressed in plants[32], an analysis of off-target effects of TadCBEa and TadDE at the transcriptomic level was conducted (Fig. 5a). In comparison to the tissue control, there were no significant changes in the total number of SNVs, C-to-U SNVs, and A-to-I SNVs (Fig. 5i–k), and the proportions of different types of SNVs remained the same in the edited plants by TadCBEa and TadDE (Fig. 5l). This suggests that the base editors did not introduce additional C-to-U and A-to-I mutations in the rice transcriptome. Further analysis of select plants edited by TadCBEa and TadDE showed that they carried more C-to-U mutations than A-to-I mutations (Fig. 5m), consistent with a similar observation about the genomic SNVs (Fig. 5d). This suggests that some of the SNVs discovered in TadCBEa and TadDE edited plants at the transcriptomic level could be due to genomic mutations, which was indeed confirmed by a further analysis (Supplementary Fig. 22). In summary, TadCBEa and TadDE did not induce significant off-target effects at both the DNA and RNA levels, suggesting they are highly specific genome editing tools in plants.

## Engineering herbicide resistance by TadDE

So far, we have demonstrated TadDE as an efficient dual base editor with high targeting specificity. With its ability to simultaneously introduce C-to-T and A-to-G base changes, TadDE holds many potential promising applications in crop improvement. To showcase some

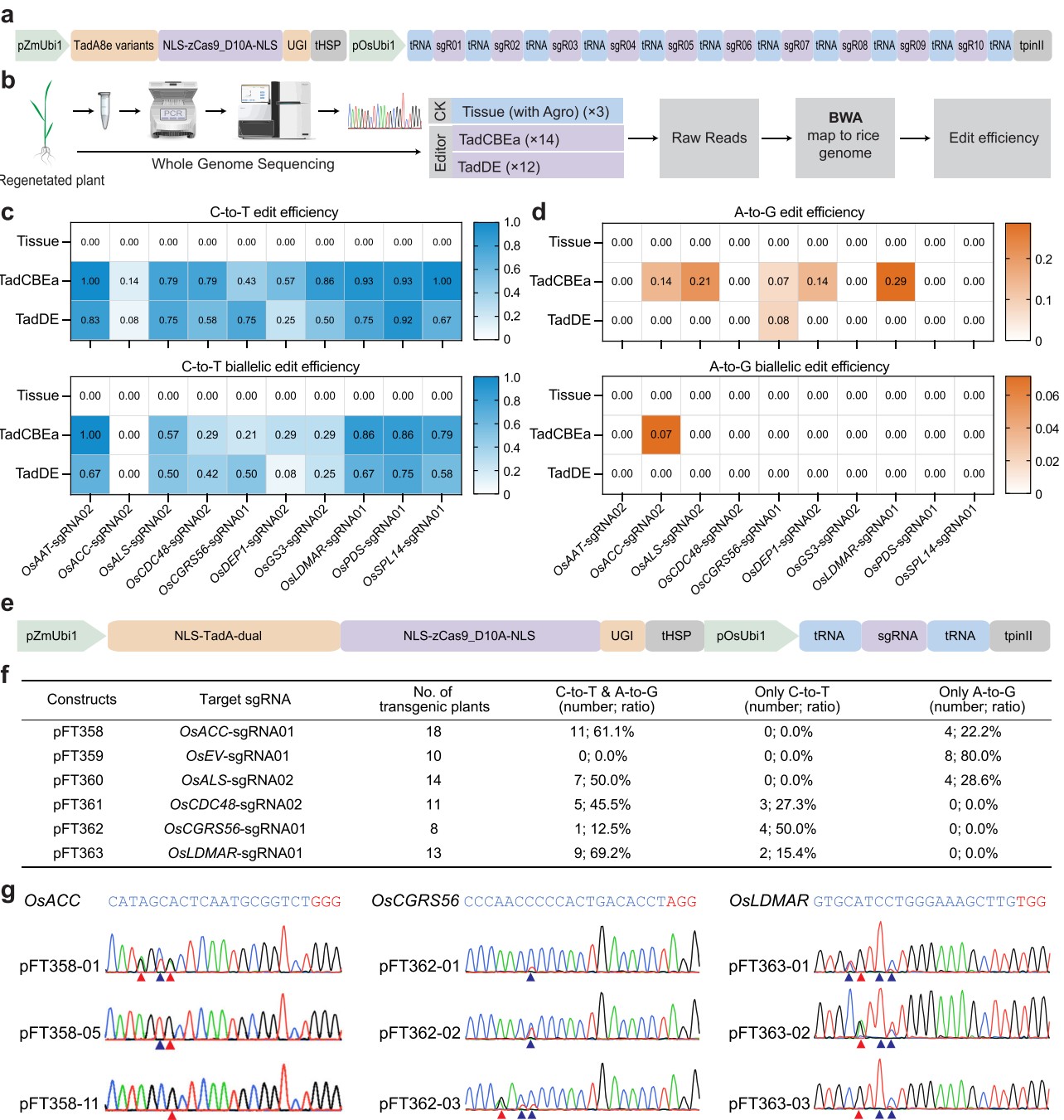

**Fig. 4 | Singular and multiplexed base editing by TadCBEa and TadDE in rice plants. a** Schematic of the multiplexed expression vector of TadCBEa and TadDE base editors targeting 10 endogenous sites with gRNA array 02. **b** Workflow for the WGS-based analysis of editing efficiency in stably transformed rice. **c** Heatmap based display of C-to-T mutation efficiency (base editing frequency > 30%) and C-to-T biallelic mutation efficiency (base editing frequency > 70%) in T0 plants based on WGS data analysis. **d** Heatmap based display of A-to-G mutation efficiency (base editing frequency > 30%) and A-to-G biallelic mutation efficiency (base editing frequency > 70%) in T0 plants based on WGS data analysis. **e** Schematic of the singular T-DNA expression vectors of TadDE. **f** Base editing type and frequency of TadA-8e derived TadDE in stable rice lines. **g** Genotyping of three selected target sites edited by TadDE singular base editors. The blue arrows indicate edited Cs, while the red arrows indicate edited As. Source data are provided as a Source Data file.

applications, we used TadDE to engineer herbicide resistance in rice by base editing of *OsALS*. Specifically, two sgRNAs (sgR17 and sgR18) targeting the DNA sequences encoding the crucial amino acids R190, G628, and G629 of OsALS were designed (Fig. 6a) and incorporated into the TadDE vector. The two TadDE constructs were then utilized for rice transformation. During the regeneration phase, 0.4 μM bispyribac-sodium was applied to select herbicide-resistant rice materials (Fig. 6b). We obtained one herbicide resistant rice plant (B1-01) with the sgR17 construct and five herbicide resistant rice plants (B2-04 to B2-08) with the sgR18 construct (Fig. 6c). Line B1-01 carried an R190H mutation (Fig. 6d), while lines B2-04 to B2-08 carried either singular or double mutations of G628E and G629S (Fig. 6e). Interestingly, all these mutations were C-to-T conversions (Supplementary Fig. 23), which could be due to herbicide-based selection of C-to-T mutations as well as absence of adenines in the editing window of TadDE (4−8 bp in the protospacers). Structural analysis revealed that these mutations all affected the catalytic core of the OsALS enzyme (Fig. 6f), which presumably prevented effective inhibition by the

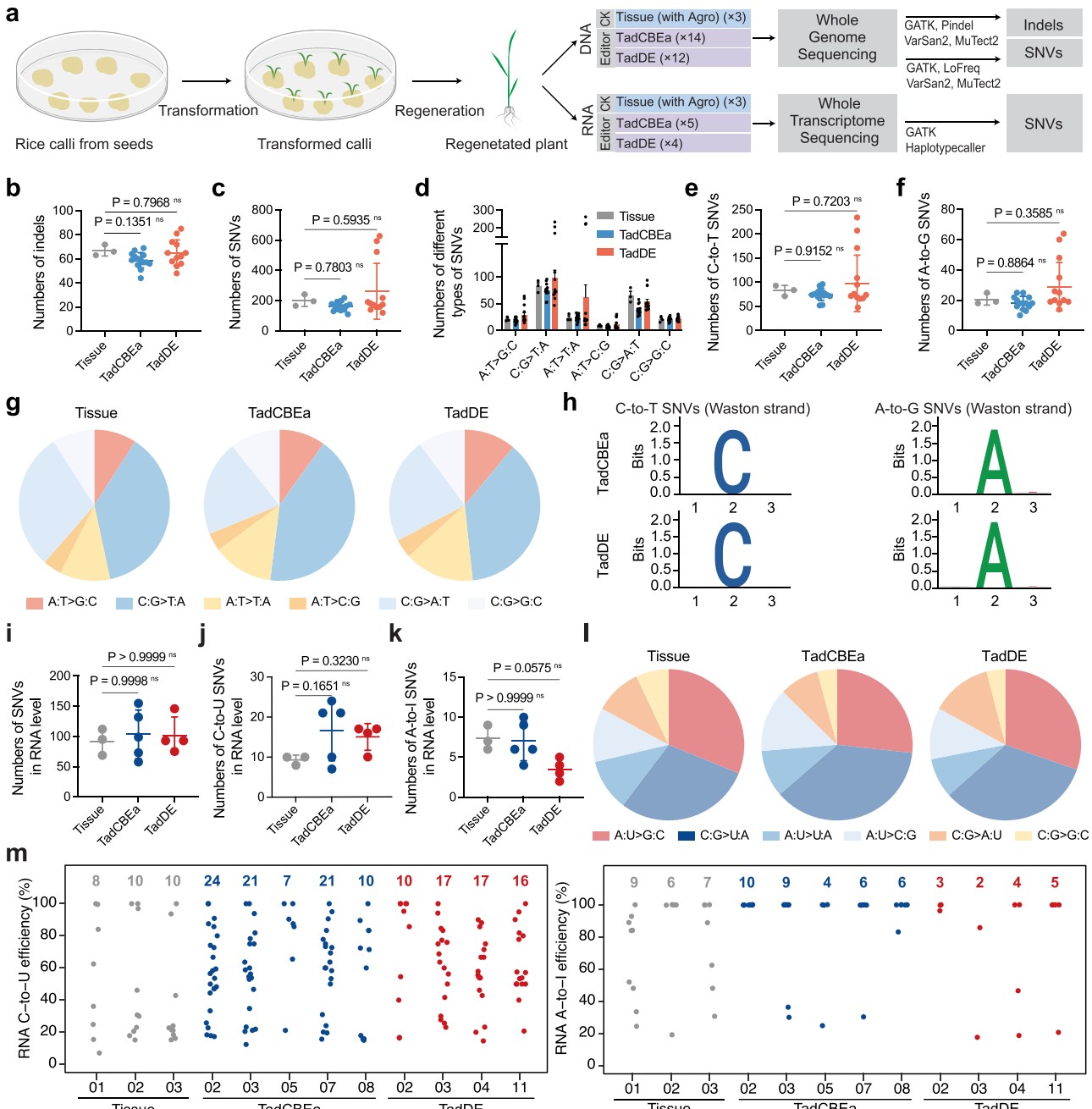

**Fig. 5 | Genome-wide and transcriptome-wide off-target assessment of TadC-BEa and TadDE in rice. a** Diagram of the experimental design of WGS and RNA-seq. **b**–**f** Numbers of indels (**b**), total SNVs (**c**), different types of SNVs (**d**), C-to-T SNVs (**e**) and A-to-G SNVs (**f**) identified in the tissue culture, TadCBEa- and TadDE-treated plants by WGS. Each dot represents the number of indels or SNVs from an individual plant. The tissue treatment includes three biological replicates ($n = 3$), the TadCBEa treatment includes fourteen biological replicates ($n = 14$), and the TadDE treatment includes fifteen biological replicates ($n = 12$). Data are presented as mean values ± SD. The statistical significance of differences was calculated using the one-way ANOVA analysis. **g** Pie charts showing the distribution of six types of genomic SNVs discovered in plants of tissue culture, TadCBEa and TadDE. **h** The sequence context of C-to-T SNVs and A-to-G SNVs in edited plants identified by WGS. Sequence conservation at positions from 1 – 3 is shown, with the mutated C or mutated A at position 2. **i**–**k** Numbers of total SNVs (**i**), C-to-U SNVs (**j**) and A-to-I SNVs (**k**) identified in the tissue culture, TadCBEa- and TadDE-treated plants by RNA-seq. Each dot represents the number of SNVs from an individual plant. The tissue treatment includes three biological replicates ($n = 3$), the TadCBEa treatment includes five biological replicates ($n = 5$), and the TadDE treatment includes four biological replicates ($n = 4$). Data are presented as mean values ± SD. The statistical significance of differences was calculated using the one-way ANOVA analysis. **l** Pie charts showing the distribution of six types of RNA SNVs induced by tissue culture, TadCBEa and TadDE by RNA-seq. **m** Analysis of RNA mutations in different T0 lines. Jitter plots show efficiency of C-to-U and A-to-I conversion mutations (y-axis) in the examined plants. Total number of modified bases is listed at the top. Each biological replicate/line is listed on the bottom. Source data are provided as a Source Data file.

herbicide. Our data suggest that TadDE is a powerful tool to engineer herbicide resistance as a gain-of-function trait in crops.

As demonstrated earlier, a single G569A base substitution in heterozygous plants results in a predicted R190H substitution in OsALS, which can be utilized for bispyribac herbicide resistance[18]. Some studies indicate that the G628E/G629S mutation confers tolerance to imazapic herbicide at 108 g ai ha⁻¹, and a triple mutant (P171F/G628E/G629S) exhibits high tolerance to all five tested herbicides,

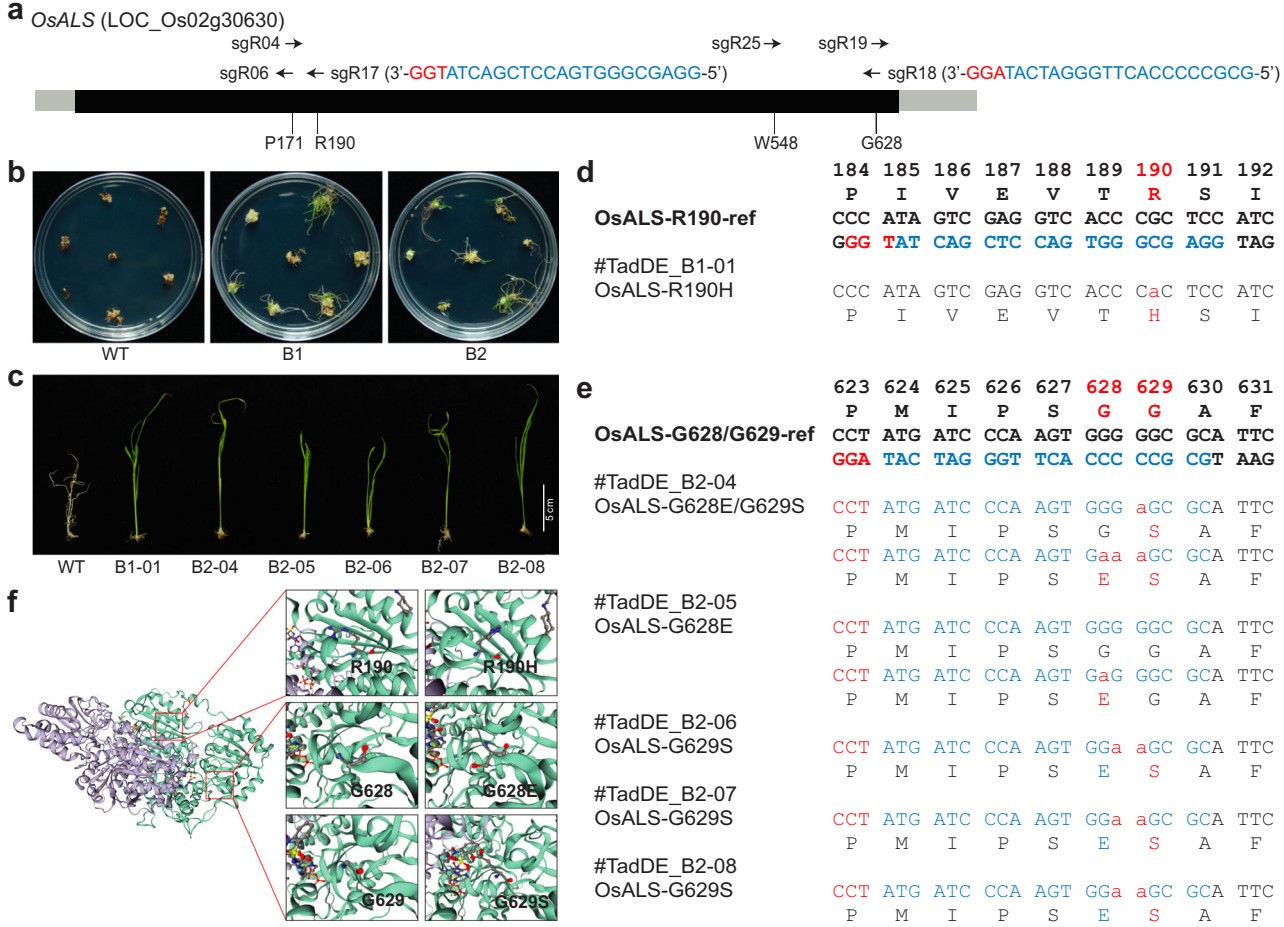

**Fig. 6 | Engineering herbicide resistance by TadDE mediated base editing of *OsALS*. a** Schematic of the target sites in *OsALS*. **b** Herbicide selection of transgenic rice calli on REIII medium with 0.4 μM bispyribac-sodium. **c** Herbicide-resistant T0 lines selected by growing the transformed and regenerated rice seedlings on the HF medium with 0.4 μM bispyribac-sodium. **d-e** Genotypes of the herbicide resistance mutations in T0 lines. **f** A structural model of OsALS protein existed in the form of a dimer, based on the structure of AtALS. The two monomers of OsALS are shown in purple and green, respectively. Key residues in the wild type (left), R190H (right 1), G628E (right 2) and G689S (right 3) are shown as sticks.

including nicosulfuron, imazapic, pyroxsulam, flucarbazone, and bispyribac[45]. In our research, we verified two herbicide-resistant loci through herbicide screening: *OsALS*-R190H and *OsALS*-G628E/G629S. Further analysis of their herbicide resistance capabilities and their application in rice breeding will be conducted through field experiments.

**Engineering micro-RNA cleavage resistant *OsSPL14* by TadDE**

To showcase another application in engineering gain-of-function in crops, we applied TadDE to edit the miRNA156 target site in *OsSPL14* with a careful designed sgRNA (*OsSPL14*-sgR05) (Fig. 7a). In the wild type plants, the transcript of *OsSPL14* is repressed by miRNA156[46]. Ideally, if we could introduce synonymous mutations at the miRNA156 target site in *OsSPL14*, the resulting edited *OsSPL14* would still encode the same protein, yet its transcript becomes resistant to miRNA156-mediated targeted cleavage. Consequently, we would expect more *OsSPL14* transcript and protein in the precisely edited plants (Fig. 7b). Among regenerated T0 rice lines, we identified six base edited plants (Fig. 7c and Supplementary Fig. 24). Notably, two lines, TadDE_B4-16 and TadDE_B4-18, carried one synonymous mutation at the miR156 target site, which was due to A-to-G base editing of the non-coding strand by TadDE (Fig. 7c). qPCR analysis in these two individual plants revealed significant up-regulation of *OsSPL14* compared to the wild type (Fig. 7d). Hence, with TadDE, we demonstrated target gene

up-regulation by mutating the miRNA target site without altering the encoded protein sequence in plants.

## Discussion

In this study, we first compared the editing performance of the TadA-8e derived CBEs in rice protoplast cells. Our data showed that the CBEs (TadCBEa, TadCBEd, TadCBEd_V106W) derived from Phage-Assisted Continuous Evolution (PACE)[40] outperformed the CBEs (eTd-CBE and Td-CBEmax) derived from structure-guided molecular engineering[39] in plants. The high base editing efficiency of TadCBEa, TadCBEd, and TadCBEd_V106W was further confirmed in tomato protoplasts. Our analyses revealed a few distinct characteristics of these TadA-8e derived CBEs that may be advantageous to the commonly used A3A_Y130F. First, these CBEs have a narrower editing window which could drastically reduce by-product base editing (Figs. 2a and 3e). Second, these CBEs showed much lower frequencies of indel by-products in tomato cells (Fig. 3d). Third, based on genome-wide off-target analysis of TadCBEa, these CBEs could be more specific than A3A_Y130F, which could induce genome-wide C-to-T off-target mutations in the rice genome[13]. Interestingly, these characteristics are consistent with the feature TadA-8e. For example, ABE8e based on TadA-8e was previously reported in rice and tomato to generate very pure A-to-G base editing without inducing indels[25,47]. The genome-wide and transcriptome-wide off-target effects of ABE8e could be detected

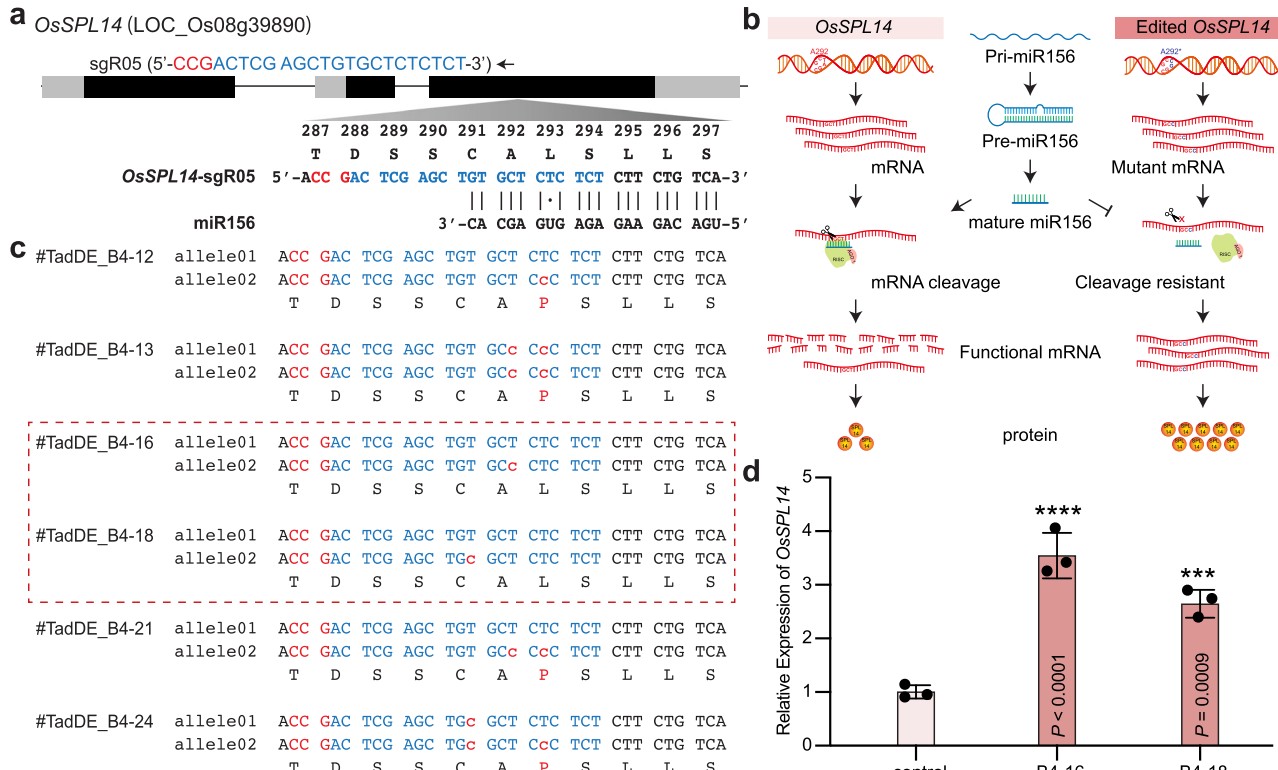

**Fig. 7 | Engineering microRNA resistant *OsSPL14* by TadDE-introduced synonymous mutations. a** Schematic of the target sites in *OsSPL14*. **b** Illustration of introducing *OsSPL14* synonymous mutations to gain resistance to microRNA-mediated degradation without altering protein sequence. **c** Genotypes of engineered microRNA resistant *OsSPL14* mutations in T0 lines. **d** Quantitative RT-PCR result of *OsSPL14* transcripts in edited T0 rice lines. Each target contains three biological replicates. Data are presented as mean ± SD The statistical significance of differences was calculated using the one-way ANOVA analysis. Source data are provided as a Source Data file.

only in high expression transgenic plants. Although we observed a few gRNA-dependent off-target mutations at the off-target sites that are almost identical to the on-target sites (e.g., only with one base mismatch), we did not detect any genome-wide gRNA-independent off-target effects in rice plants expressing TadCBEa and TadDE, which could be due to a counter-selection effect against high expression of the deaminase in the multiplexed editing constructs.

A significant innovation of this study is the demonstration of a latest dual base editor, TadDE in plants. Previously, simultaneous C-to-T and A-to-G base editing was achieved by recruiting a cytidine deaminase and an adenosine deaminase simultaneously via different approaches. For example, the simultaneous wide-editing induced by a single system (SWISS) utilized engineered sgRNA scaffolds to recruit cytidine deaminases and adenosine deaminases via different RNA aptamer-RNA binding protein interactions[13,15]. However, the SWISS systems could only confer cytosine base editing at one target site and adenine base editing at the other[36]. It was through direction fusion of both cytidine deaminase and adenine deaminase to the Cas9 nickase, simultaneous C-to-T and A-to-G base editing was achieved at the same target site[35]. These saturated targeted endogenous mutagenesis editors (STEMEs) were shown to direct targeted protein evolution of *OsACC* to engineer an herbicide resistance trait in rice[35]. The STEME systems mostly induce C-to-T base editing than simultaneous C-to-T and A-to-G base editing, which could be partly attributed to the use of a low efficiency adenine deaminase[35]. A recent study adopted a similar strategy of the SWISS system and took one more step further to install two distinct RNA aptamers on the sgRNA scaffold for the recruitment of cytidine deaminase and adenine deaminase simultaneously to the same target site[37]. This multiplexed orthogonal base editor (MoBE) system greatly outperformed the STEME system by inducing simultaneous C-to-T and A-to-G mutations at the same target sites efficiently.

The authors also demonstrate the use of this dual base editing system to engineer herbicide resistance by targeting *OsACC* with a sgRNA library[37]. These studies demonstrated the usefulness of dual base editors for targeted random mutagenesis in plants.

We found TadDE is a highly efficient dual base editor as long as there are editable cytosines and adenines in the editing window. Impressively, no significant off-target effects of TadDE were found based on genome-wide and transcriptome-wide analyses. This suggests that this latest dual base editor is not only simple and efficient but also highly specific. Construction of the MoBE dual base editing system is complicated as it requires the expression of four protein effectors (e.g., two different deaminases and two different RNA binding proteins) to achieve the desired dual base editing[37]. By contrast, the TadDE that we demonstrated here is very simple, only requiring a small engineered TadA-8e variant that is fused to the N-terminus of nCas9. Hence, construction of the TadDE base editors is much more streamlined as it retains the same configuration with typical CBEs and ABEs. This simplicity will further facilitate DNA-free ribonucleoprotein (RNP) delivery of base editors to achieve transgene-free base editing in crops.

We demonstrated two gain-of-function trait engineering applications using TadDE, one focusing on engineering of herbicide resistance trait and the other focusing on the up-regulation of an endogenous gene by inhibiting its negative regulation by a miRNA. CRISPR-based genome editors have been commonly used for gene knockout by targeting the coding sequences[15,34,48] or knockdown by targeting the promoters[16,49,50]. However, limited options are available to achieve gene up-regulation or knock-up by genome editing in plants. Previously, such outcomes were obtained by editing upstream open reading frames (uORFs)[19,51]. More recently, it was demonstrated that up-regulation of a gene expression could also be achieved by editing 3' UTR-embedded inhibitory regions[52]. Notably, our engineering of

miRNA-resistance transcripts by installing synonymous mutations represents a third strategy to over-express of an endogenous gene in plants. This strategy would be widely applicable as many genes in plants are control by miRNAs. Although we mutated miRNA target sites by using a base editor here, it is conceivable that genome editing tools with higher precision such as prime editors may be needed to achieve this effect more efficiently[34].

Our study also reported TadDE-NG, which should enable targeting at relaxed NG PAM sites, hence greatly expanding the target range of this dual base editor. Expanding the targeting scope of base editors can also be achieved with the use of CRISPR-Cas12a base editors that recognize A/T-rich PAMs, as recently demonstrated in plants[53–55]. The previously demonstrated Cas12a CBEs had relatively poor base editing activity. It is worth testing whether the engineered TadA-8e based cytidine deaminases as demonstrated in our study would augment C-to-T base editing efficiency when fused to the Cas12a protein. It would be even more interesting to use the TadA-8e deaminase in TadDE to engineer Cas12a dual base editors. After all, due to lack of complexity of the CRISPR RNA (crRNA) of Cas12a, it is nearly impossible to engineer a Cas12a dual base editor via engineering of the crRNA scaffold as with the MoBE system[37]. Hence, the same strategy applied in TadDE represents a most straightforward approach to engineer efficient Cas12a dual base editors, which would enable efficient targeted mutagenesis of A-T rich promoter regions to engineer quantitative traits in crops. Moreover, we envision that fusing these small TadA-8e variants to a small Cas12 protein such as CasΦ (Cas12j2) may help engineer next-generation compact dual base editors of small sizes that are compatible to viral delivery in plants[56,57].

In summary, we developed and demonstrated TadCBEs and TadDE dual base editors for precise plant genome editing. These base editing tools are of high efficiency, purity, and specificity. We expect that further applications and improvement of these tools would unleash the great potential of the ever-expanding base editing toolbox toward innovative crop engineering.

## Methods

### Vector construction and target genes
The vectors were constructed based on the backbone of pTX1500. The DNA sequences encoding chosen TadA variants were rice codon optimized, synthesized, and constructed into pUSP-Empty by Gibson Assembly. For assembly of nCas9 and deaminase expression cassettes, the promoter, the deaminase, the nCas9 and the terminator elements were cloned in pTSWA at BsaI sites by Golden Gate Assembly. For assembly of T-DNA expression vectors without target sgRNAs, pTSWA-derived Module A and Module B which has the sgRNA scaffold and tRNA fragment were digested and ligated into pTX1500 at AarI sites by Golden Gate Assembly. For constructing single site base editor, the protospacer sequences were synthesized as oligos by Sangon Biotech. The oligos were annealed and cloned into T-DNA backbone vector. For generating multiplexed base editors, sgRNA arrays were amplified and cloned into T-DNA backbone vector at BsaI by standard Golden Gate reaction. These vectors were confirmed by SacI digestion and Sanger sequencing.

For base editing in tomato, the pYPQ265E2 (Addgene # 164719) was used for generating attL1-attL5 Cas12a entry clones. The four TadA variants and 2xUGI were synthesized as gBlocks® by IDT. Firstly, the UGI in pYPQ265E2 was replaced by 2xUGI using the NEBuilder HiFi DNA Assembly Cloning Kit (New England Biolabs®) to generate A3A-Y130F-nzCas9-2xUGI. Then, the TadCBEa, TadCBEd, TadCBEd-V10W, and TadDE were assembled with the purified PCR nzCas9-2xUGI fragment to generate pYPQ265E6 (TadCBEa-nzCas9-2xUGI), pYPQ265E7 (TadCBEd-nzCas9-2xUGI), pYPQ265E8 (TadCBEd-V106W-nzCas9-2xUGI), pYPQ265E9 (TadDE-nzCas9-2xUGI), respectively, using the HiFi cloning method. As for the target sites, protospacers were synthesized as single strand oligos by GeneWiz. After phosphorylation and annealing, the four paired of oligos were ligated into pYPQ131B (Addgene # 69281),

pYPQ132B (Addgene # 69282), pYPQ133B (Addgene # 69283) and pYPQ134B (Addgene # 179216) at the BsmBI site with Instant Sticky-end Ligase Master Mix (New England Biolabs®), respectively. Then, the four sgRNA cassettes were assembled into the attL5-attL2 vector pYPQ144 (Addgene # 69296) using Golden Gate assembly[58]. T-DNA expression vector was assembled by the three-way Gateway LR reaction with an attL1-attR5 cas9 base editor entry clone, an attL5-attL2 sgRNA entry clone, and an attR1-attR2 destination vector pCGS710.

All vectors used in this study were listed in Supplementary Table 1. All target sites were listed in Supplementary Table 2. The oligos used in this study were summarized in Supplementary Data 1.

### Plant material and growth condition
The Japonica cultivar Nipponbare of rice (Oryza sativa) was used in this study. For rice protoplast transformation, seedlings were cultivated on 1/2 MS solid medium in the dark at 28 °C for 11 days. For stable rice transformation, sterilized seeds were placed on N6-D solid medium to induce calli for 7 days in the light at 32 °C.

The tomato (Solanum lycopersicum) cultivar M82 was utilized for the tomato protoplast assay. Tomato plants were grown in 1/2 MS media at 25 °C with a photoperiod of 12 h light and 12 h dark. Protoplast isolation was conducted using 10 day-old seedlings.

### Rice protoplast transformation and mutation analysis
Rice protoplast isolation and transformation was carried out in accordance with previously published protocols[59–61]. Briefly, 11 day old healthy rice leaves were cut into 1.0–2.0 mm strips and transferred into the enzyme solution followed by vacuum-infiltration for 30 min, and then incubated at 25 °C in the dark at 70–80 rpm. The rice cells and enzyme solution mixture were filtered by 40 μm cell strainer after digestion for 8 h. Rice cells in the filtrate were collected into 50 mL centrifuge tube and washed twice with W5 buffer. Then protoplasts were examined and counted under a microscope. The final protoplast concentration was adjusted to $2 \times 10^6$/mL. For protoplast transformation, 30 μg plasmid DNA (1 μg/μL; prepared by QIAGEN Plasmid Midi Kit) in 30 μL MMG solution was used to transform 200 μL protoplasts by gently mixing with 230 μL 40% PEG transformation buffer. After incubation for 30 min in dark, the reactions were stopped by adding 1000 μL W5 washing buffer. The protoplasts were centrifuged and transferred into 12-well culture plates and left at 32 °C in dark for 48 h before collection.

The collected rice protoplasts were used for DNA extraction using the CTAB method[62]. With the protoplast DNA as template, ~250 bp sequence containing the target sites were PCR amplified using barcoded primers. Then purified PCR products were sent to the Novogene for NGS sequencing and the clean data were analyzed by CRISPRMatch software[63,64].

### Tomato protoplast transformation and mutation analysis
The tomato protoplasts were isolated and transformed according to the previously described method[65]. In brief, the cotyledons were cut from 10 day-old tomato seedlings and subjected to enzyme solution digestion for around 8 h at 28 °C in the dark at 65 rpm. The digested cells were filtered through a 75 μm cell strainer and washed with W5 buffer. Then protoplast suspension was subjected to centrifugation for 10 min. The resulting pellet of protoplasts was resuspended by 0.55 M sucrose and gently overlaid with W5 solution. Following a centrifugation step for 30 min, the protoplasts were carefully collected from the interface between the sucrose and W5 layers by pipette. The protoplasts were then washed with W5 solution twice.

For the transformation of tomato protoplasts, 20 μg of plasmid DNA (1000 ng/μL) was added to 200 μL of the protoplast suspension (density $5 \times 10^5$/mL) in MMG solution and mixed gently. Subsequently, 220 μL of 40% PEG solution was added to the plasmid and cell mixture, and again mixed gently. This mixture was then left to incubate for

20 min at room temperature. The transformation process was stopped by adding 900 µL of W5 buffer. Following this, the protoplasts were collected via centrifugation and transferred to 12-well culture plates, which were then incubated at 30 °C in the dark for 60 h. After incubation, the protoplasts were collected through centrifugation, lysed, and PCR amplified with barcoded primers using Phire Plant Direct PCR Mix (ThermoFisher). The PCR amplicons were purified and pooled for NGS. The NGS data was analyzed by CRISPRMatch software[63].

## Rice stable transformation

Agrobacterium-mediated rice transformation was conducted by following a modified protocol[66,67]. In brief, calli derived from rice seeds were induced in a growth chamber for 7 days at 32 °C under light. T-DNA expression vectors were introduced into Agrobacterium tumefaciens strain EHA105. The transformed Agrobacterium cells were collected and resuspended in liquid AAM-AS medium (OD600 = 0.1) containing 100 µM acetosyringone. Following this, rice calli were immersed in AAM medium containing Agrobacterium. After a duration of 2 min and 30 s, the infiltration solution was then discarded. Following a 3 day co-culture of Agrobacterium and calli in N6-AS medium, the calli were washed with sterilized water and transferred to N6-S medium containing 200 mg/L Timentin and 50 mg/L Hygromycin for a 2 week period. Subsequently, the calli were moved to REIII medium supplemented with 200 mg/L Timentin and 50 mg/L Hygromycin for an additional 2 weeks, and resistant calli were transferred to fresh REIII every 2 weeks until regenerated plants were obtained. Herbicide-resistant rice lines were screened using REIII medium containing 0.4 µM bispyribac-sodium.

## Mutation detection and analysis of T0 rice lines

DNA from T0 plants was extracted with the CTAB method[62]. With the plant DNA as template, PCR products were amplified, and Sanger sequenced to preliminarily confirm the result of base editing in rice plants. Then, 10 ~ 15 lines for each base editing construct were proceeded for WGS and 3 ~ 5 lines were selected for RNA-seq.

## WGS data analysis

The analysis of the whole genome sequencing (WGS) was conducted following an established protocol[25,43,68] with some minor revisions. Adapter sequences were trimmed from the raw sequencing reads using the SKEWER program (version 0.2.2)[69]. The resulting cleaned reads were then aligned to the rice reference genome (MSU7), accessible through http://rice.uga.edu/, utilizing the BWA mem tool (version 0.7.17)[70]. To sift out reads that were non-uniquely mapped, both Picard tools and Samtools (version 1.9)[71] were used. The Genome Analysis Toolkit (GATK, version 3.8)[72] was implemented for the realignment of reads in regions flanking indels. For the detection of whole genome SNVs and indels, a combination of analytical tools was employed which are LoFreq (version 2.1.2)[73], Mutect2[74], VarScan2 (version 2.4.3)[75] and Pindel (v. 0.2)[76]. Intersection of identified SNVs and indels was performed using Bedtools (version 2.27.1)[77]. In addition to these steps, Cas-OFFinder (version 2.4)[44] was used to predict potential off-target regions, allowing for a mismatch leniency of up to three nucleotides. For the processing and analytical tasks that followed, programming languages Python and R were applied to handle and analyze the data efficiently. The spatial arrangement of mutations across the genome was visualized using Circos (version 0.69)[78].

## RNA-seq data analysis

The relevant analytical methods have already been established in previous research[47,61]. In brief, clean reads were mapped to the rice reference genome (MSU7) utilizing the Hisat2 application (version 2.2.0)[79]. The subsequent processing of BAM files, which involved sorting and flagging of duplicate reads, was managed using Picard. The process proceeded with tackling reads over splice junctions, realigning them locally, and calling variants, which were executed via GATK

(version 3.8) functionalities including SplitNCigarReads, IndelRealigner, and HaplotypeCaller. Attention was concentrated on pinpointing SNVs within the primary chromosomal range of Chr1 to Chr12, filtering putative RNA SNVs using the VariantFiltration tool to identify only those of high confidence. Comprehensive data handling and analytical procedures were carried out leveraging the capabilities of Python and R to ensure a thorough and structured analysis of the genome data. Sequences of three base pairs flanking each single nucleotide variant (SNV, A-to-G or C-to-T) were retrieved from the reference genome. These sequences were then analyzed with WebLogo3 (http://weblogo.threeplusone.com/)[80] to create a graphical representation of the sequence motif.

## qRT-PCR analysis

Total RNA was extracted from plant leaves using FastPure Universal Plant Total RNA Isolation Kit (Vazyme). The extracted RNA was reverse transcribed using the HiScript III RT SuperMix for qPCR kit (Vazyme). qRT-PCR was done using ChamQ Universal SYBR qPCR Master Mix (Vazyme). Each qRT-PCR assay was repeated at least three times with three independent RNA preparations, and the rice Actin 1 (OsActin1) gene was used as a reference.

## Statistics & Reproducibility

In this study, no statistical method was used to predetermine the sample size. No data were excluded from the analyses. All samples were randomly distributed across all replicates. For each experiment, the treatments were compared to a control treatment without any prior knowledge of whether the experimental variables being altered would have a positive or negative impact on the results.

## Reporting summary

Further information on research design is available in the Nature Portfolio Reporting Summary linked to this article.

## Data availability

Key vectors used in this study have been deposited to Addgene: pGEL850 (expression TadCBEa in rice cells, Addgene # 214250); pGEL851 (expression TadCBEd in rice cells, Addgene # 214251); pGEL852 (expression TadCBEd_V106W in rice cells, Addgene # 214252); pGEL853 (expression TadDE in rice cells, Addgene # 214253); pGEL854 (expression eTd-CBE in rice cells, Addgene # 214254); pGEL855 (expression Td_CBEmax in rice cells, Addgene # 214255); pGEL856 (expression TadDE-NG in rice cells, Addgene # 214256); pYPQ265E6 (TadCBEa-nzCas9-2xUGI, Addgene # 213469); pYPQ265E7 (TadCBEd-nzCas9-2xUGI, Addgene # 213470); pYPQ265E8 (TadCBEd-V106W-nzCas9-2xUGI, Addgene # 213471); pYPQ265E9 (TadDE-nzCas9-2xUGI, Addgene # 213472). The plasmids mentioned above have been submitted to the Addgene (https://www.addgene.org/browse/article/28243932/). The NGS data generated in this study have been deposited in the National Center for Biotechnology Information (NCBI) database under Sequence Read Archive (SRA) with the BioProject ID PRJNA1054338. Similarly, the WGS and RNA-seq data have been deposited in the NCBI database under SRA with the BioProject ID PRJNA1059495. Source data are provided with this paper.

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

## Acknowledgements

We thank Colby Starker and Daniel Voytas for sharing the pCGS710 vector. This research was supported by the National Key Research and Development Program of China (award no. 2023YFD1202900) to X.T. and Y.Z., the National Natural Science Foundation of China (award no. 32270433, 32101205, 32072045) to Y.Z., X.T., X.Z., the Natural Science Foundation of Sichuan Province (award no. 2022NSFSC0143) to X.T. It is also supported by the NSF Plant Genome Research Program (IOS-2029889 and IOS-2132693) to Y.Q.

## Author contributions

Y.Z. proposed the project. Y.Z. and Y.Q. conceived and designed the experiments. X.T. designed the backbone vector. T.F. and S. Liu constructed all the T-DNA base editing vectors for rice. T.F. performed rice protoplast transformation. T.F., Y.W., and T.Z. analyzed the mutation frequencies in rice protoplasts. Y.C. constructed the Gateway entry vectors deposited to Addgene and the T-DNA base editing vectors for tomato. Y.C. conducted tomato protoplast transformation experiments and data analysis. T.F., S. Liu, S. Liao, Y.H., and X.Z. performed rice stable transformation. T.F. prepared rice seedling samples for Sanger sequencing, WGS, and RNA-seq. T.F., Y.W. and T.Z. analyzed the editing efficiency and specificity in rice stable T0 lines by examining WGS and RNA-seq data. Y.Z., Y.Q., T.F., Y.W., T.Z., and X.T. organized the main figures, Supplementary Figs. and supplementary table of the article. Y.Z., Y.Q., T.Z., T.F., and X.T. analyzed the data and wrote the paper with input from other authors. All authors read and approved the final version of the manuscript.

## Competing interests

The authors declare no competing interests.
