## [Peer Review File · Nature Communications]

High performance TadA-8e derived cytosine and dual base editors with undetectable off-target effects in plantsEditorial Note: This manuscript has been previously reviewed at another journal that is not operating a transparent peer review scheme. This document only contains reviewer comments and rebuttal letters for versions considered at *Nature Communications*. Mentions of the other journal have been redacted.

REVIEWER COMMENTS

Reviewer #1 (Remarks to the Author):

Both original reviewers critiqued that the work is not significantly novel enough to justify publication [Redacted]. The authors respond that the work is novel due to 1) adaptation and optimization of plant systems, 2) it addresses unique challenges in plants, 3) there are novel insights and applications and 4) practical application for crop improvement. Overall, I am not convinced by these arguments. I will go through point by point.

1: True the editors reported here were adapted for plant expression. However this was all using highly standard parts (maize ubi promoter) and transformation methods (PEG and agrobacterium). The base editors were not optimized in any sense of the word. A screen was performed in protoplasts and stable plants. Off target effects were not minimized. They were evaluated. While it is true that some CRISPR and other DNA repair technologies are difficult to port from one kingdom to another (HDR, prime editing), this has not been the case for CRISPR base editors. They work essentially the same in mammalian and plant systems. There is an exhaustive list of various deaminase enzymes being tested in various plant species. This manuscript reports yet another.

2: This is really not much different than point 1. I do not follow the arguments that efficacy can vary due to "genomic architecture, cell biology and regeneration capabilities". No references are provided to support the arguments.

3. As noted in the text, other dual base editor CRISPR systems have been reported in plants. These use a nickase Cas9 fused with both a cytidine and adenine deaminase proteins. So such a dual-editing system is not novel in plants. The enzyme is novel (in plants), but as noted above, there are many different types of deaminases reported. The smaller molecular weight is not a convincing argument. The deaminase domains are relatively small (150-230 AAs) compared to cas9 (1,380) and the delivery methods used were not limited by cargo size. There is no way for the smaller molecular weight to be "showcased" in this report.

4. Targeting the ALS gene in plants is nothing new. This has been one of the standard genome editing targets since the beginning. SPL14/miR156 is known, but targeting it with base editors is unique as far as I can tell. While gene upregulation was observed, this is a far way away from being a practical application for crop improvement.

Regarding the lethality comment from R1, the authors respond that the combination of low regeneration rates from the multiplex vectors and high editing rates in the single targets indicate lethality due to multiplexing. This is the definition of hand waving. Transformation and regeneration rates are well known in the field to be highly variable. If someone wants to claim anything regarding transformation efficiency, multiple transformation experiments and controls are required. But it appears these claims are based on single experiments without controls. I do not know how one could claim reduced transformation efficiency based on this experimental setup. Indeed, this experiment was not designed to address this question. I do not understand how the results in supplemental table 3 support the lethality argument. The low frequencies appear to occur across the board, regardless of dual, cytosine or adenine base editor. This observation could very well be due to any number of technical issues. The idea that high rates of base editing lead to lethality is a rather favorable way to interpret these ambiguous results.

I agree with parts of the response regarding the off-target effects. I think R2 is unreasonable to think that off-targets are not relevant for plants. True, one can perform backcrossing. But I agree with the authors that extensive backcrossing is a significant burden for plant breeders. At the same time, the response really oversells what has been reported. Social responsibility?

Reviewer #2 (Remarks to the Author):

This study introduces high-performance cytosine and dual base editors derived from TadA-8e, showing undetectable off-target effects in plants. The research focuses on engineering TadA-8e to achieve both cytosine and dual base editing with high specificity and efficiency in rice and tomato cells. The development of these editors, particularly TadCBEa, TadCBEd, TadCBEd_V106W, and TadDE, is significant for their high editing purity, narrow editing window, and the ability to perform multiplexed base editing in transgenic rice plants without detectable off-target effects through whole genome and transcriptome sequencing.

While base editors have demonstrated efficacy in mammalian systems first, their effectiveness in plant systems cannot be assumed solely based on their performance in mammalian systems. Consequently, systematic testing and optimization of base editors in plant systems are crucial for advancing research on plant functional genomics, crop genetic improvement, and the development of new germplasm. The authors' research contributes valuable insights for researchers involved in plant gene editing and crop breeding endeavors.

Reviewer #1 (Remarks to the Author):

Both original reviewers critiqued that the work is not significantly novel enough to justify publication [in N. plants]. The authors respond that the work is novel due to 1) adaptation and optimization of plant systems, 2) it addresses unique challenges in plants, 3) there are novel insights and applications and 4) practical application for crop improvement. Overall, I am not convinced by these arguments. I will go through point by point.

Authors' response: Thanks to Reviewer #1 for your feedback on the manuscript. We regret that our previous response was not fully recognized by you. Please refer to the three major points mentioned above regarding the novelty and significance of this work.

Before getting to our point-to-point response, we would like to elaborate the significance and novelty of this study that comprehensive covers all three major areas/questions of developing and demonstrating genome editing tools: 1) Are the tools efficient and among them which one has the highest efficiency and hence is more likely to be adopted by the research community? 2) What is the off-targeting effects of the new genome editing tools, at both DNA and RNA levels? 3) What are potentially useful applications of the tools, with showcasing some exciting examples that have not been done before?

First, we did a comprehensive comparison of multiple TadA8e-derived CBEs, developed by different groups, as three Nature Biotech papers. As our data showed (Figure 1), the CBEs (TadCBEa, TadCBEd, and tadCBEd_v106w) developed using protein evolution by David Liu's group significantly outperformed those CBEs (dTd-CBE and Td-CBE_{max}). This knowledge is novel and significant to report to the research community, because it pointed to certain high-efficiency TadA8e-based CBEs to use in plants and beyond. Furthermore, since the field has been using many CBEs based on different cytidine deaminases, it is important to benchmark these new CBEs with the commonly used one. We chose to benchmark against A3A-Y130F as this CBE has shown high and robust C-to-T base editing activity across plant species such as rice¹, tomato², and poplar³. This comparison showed the top performing TadA8e-derived CBEs had comparable editing efficiency to A3A-Y130F-based CBE (Figure 1), yet they (just like TadA8e-derived ABEs) had narrower editing window than A3A-Y130F in both rice (Figure

2) and tomato (Figure 3). Again, this information will direct users to choose appropriate CBEs in their tailored applications. The high editing efficiency of the representative new CBE, TadCBEa, was further confirmed in stably transformed rice lines under a multiplexed editing setting (Figure 4).

Second, this study also explored a new type of dual base editor, TadDE, with close comparison to both CBEs and ABEs. The side-by-side experiments in monocot (rice) and dicot (tomato) cells allowed us to carefully benchmark its editing efficiency and windows (Figure 1 to Figure 3). We also showcased two applications of this TadDE for crop engineering. The first case of engineering of herbicide resistance is a relatively ordinary example as many such cases have been reported previously by using CBEs or ABEs alone. However, this experiment nevertheless identified a few new OsALS allele that confer herbicide resistance (Figure 6). The second case that we showed however is extraordinary. To our knowledge, this is the first demonstration of engineering gain-of-function for the target gene (OsSPL14) by editing the miRNA (OsmiR156) target site without altering the protein coding sequence. This application alone has a high novelty.

Third, we comprehensively analyzed potential off-target effects of multiplexed genome editing using the new CBE (TadCBEa) and the new dual base editor (TadDE) in rice. Previously, our group has published many genome-wide off-target analysis studies in plants for different genome editors, such as Cas9 and Cas12a in rice ⁴, a series of CBEs in rice ⁵ and tomato ⁶, TadA-8e in rice ⁷ and in tomato ⁸, PAM-relaxed genome editors in rice ⁹, and Cas12a-mediated multiplexed genome editing in rice assessing what happens when the genome endures multiple DNA double strand breaks (DSBs) ¹⁰. The current USDA-APHIS regulation only allows one targeted edit in a crop to be de-regulated. If the base editor introduces a second edit (or many edits) elsewhere in the genome due to off-targeting, the crop simply cannot be de-regulated. So, it is not the matter of using crossing to clean up the background or not. Rather, we need very precise base editors in the first place to create crops that will be de-regulated, at least in the USA. That's why we consider it important to develop highly specific genome editing tools with minimized off-target effects. Previously, off-target effects of base editors haven't been closely studied when used in a multiplexed setting, whether in mammalian cells or in plants. Here, we conducted genome-wide and transcriptome-wide analyses on these two new base editors

in a highly multiplexed genome editing setting, editing 10 target sites simultaneously. This not only means that we will look for off-target effects by 10 independently sgRNAs, but also means we can examine the genome integrity when it sustained at least 10 simultaneous DNA nicking. This represents the first ever genome and transcriptome-wide analyses of off-target effects of these two high performance base editors in a whole organism. That makes Nature Communications a suitable journal to publish our work. Anyway, we really appreciate the sentiments from both Reviewers here on the importance of editing specificity and study of off-target effects.

The above-mentioned three aspects could be reported in three separate papers. However, we believe it is far more meaningful to present them together in a comprehensive, solid, and interesting article rather than reporting them in separate stories. Considering the added weight and novelty of these three coherent parts, we believe that this paper has significant novelty that merits the publication at the high level journal like Nature Communications. The following are point-to-point responses to your comments.

1: True the editors reported here were adapted for plant expression. However, this was all using highly standard parts (maize ubi promoter) and transformation methods (PEG and agrobacterium). The base editors were not optimized in any sense of the word. A screen was performed in protoplasts and stable plants. Off target effects were not minimized. They were evaluated. While it is true that some CRISPR and other DNA repair technologies are difficult to port from one kingdom to another (HDR, prime editing), this has not been the case for CRISPR base editors. They work essentially the same in mammalian and plant systems. There is an exhaustive list of various deaminase enzymes being tested in various plant species. This manuscript reports yet another.

Authors' response: Thank you for your review and feedback. We agree with this Reviewer #1 that our work is not centering on base editor optimization. Rather, we compared the newly reported and very promising TadA-8e derived CBEs and TadDE (from three back-to-back studies) in the same vector backbones and expression systems. We believe this is the best way to compare them. We also agree with the Reviewer #1 that often times there may not be sufficient DNA repair difference to account for any difference in genome

editing outcomes in mammal's vs in plants. However, we have to examine these in experiments. Hence, we chose to conduct close comparison and benchmarking these systems, which is the reason that our work is unique and novel. For example, we indeed found that not all base editors that performed well in animal cells could effectively edit in plant cells. eTd-CBE and Td-CBEmax showed lower base editing efficiency in rice cells compared to their performance in animal cells. Previously, fusing of a select DNA-binding domain (DBD) was shown to enhance base editing outcomes in mammals and plants ^{11, 12}. After our manuscript submission, we also optimized the base editors by fusing a proven DBD between the deaminase and nCas9 protein to explore whether DBD would enhance base editing efficiency. However, the results indicated that DBD did not improve the base editing efficiency of TadCBEa and TadDE (supporting figure 1). Again, this tells us that we cannot simply assume everything demonstrated in mammalian systems will automatically work in plants.

We will further address your three key concerns one by one as follows.

1) "The base editors were not optimized in any sense of the word."

Authors' response: We must respectfully disagree with your assertion that "no optimization was performed on the base editors." At the outset of our research, based on experiences from different plant laboratories and our own work on base editing in 2021¹, considering the focus on developing new base editors and the competitive nature of this work, we conducted preliminary screening of base editors. We selected the optimal backbone vector in plants to test different base editing domains. Through a rational analysis of three back-to-back published studies on mammalian TadCBEs and TadDE ^{13, 14, 15}, we ultimately selected seven excellent deaminase sequences from two papers for evaluating the base editor activity in this manuscript. Additionally, based on recent work on improving editing efficiency through fusion with Cas protein DBD¹², we have just completed the optimization of TadCBEa and TadDE. We evaluated the efficiency of TadCBEa and TadDE at 10 rice sites in rice protoplasts within array 02. The results indicate that fusion with DBD did not effectively enhance the editing efficiency of

TadCBEa and TadDE (supporting figure 1). So, your statement on “not optimized in any sense of the word” is too strong to summarize this work.

2) “Off target effects were not minimized.”

Authors’ response: We don’t disagree with this statement. However, it is important to assess off-target effects of any new base editors as our previous studies clearly demonstrated that different base editors can have different levels of off-target effects. For example, the A3A-Y130F CBE showed detectable off-target effects in rice¹, while the TadCBEa and TadDE did not as we found in this study. As we elaborated in our opening remarks of this responses letter, it is important to have base editors have minimized off-target effects, which can be done by selecting the right deaminases from different origins or through protein engineering.

3) "While it is true that some CRISPR and other DNA repair technologies are difficult to port from one kingdom to another (HDR, prime editing), this has not been the case for CRISPR base editors."

Authors’ response: Thank you for acknowledging the validity of the statement regarding the difficulty in porting some CRISPR and other DNA repair technologies from one kingdom to another (HDR, prime editing). Differences between species are applicable to CRISPR base editors. Some following examples can further illustrate this issue: (1) Before the development of ABE8e, all ABEs could effectively edit in animal cells but not in plant cells^{16, 17, 18, 19}; (2) Base editors based on CRISPR-Cas12a can effectively edit in animal cells, but their editing activity in plants is extremely low^{20, 21, 22, 23}; (3) APOBEC3A exhibits higher editing activity in plants, whereas APOBEC1 is most effective in animal cells^{1, 24}; (4) Novel base editors such as AKBE^{25, 26, 27} and CGBE^{28, 29, 30, 31} can edit in animal cells, but their editing activity is extremely low in plant cells; (5) Multiple laboratories, including ours, have compared various CBEs, and their editing efficiency and specificity in plants do not consistently match their performance in animal cells^{1, 32, 33, 34}.

Please further refer to our reponses to your point #2 below for our thoughts on why there are different editing outcomes in mammalian systems and in plants.

Supporting Figure 1. The editing efficiencies of TadCBEa and TadDE fused with DBD. **A** The schematic representation of the vector; **B** The C-to-T editing efficiencies of TadCBEa

and TadDE fused with DBD; **C** The A-to-G editing efficiencies of TadCBEa and TadDE fused with DBD; **D** The Indel editing efficiencies of TadCBEa and TadDE fused with DBD.

2: This is really not much different than point 1. I do not follow the arguments that efficacy can vary due to “genomic architecture, cell biology and regeneration capabilities”. No references are provided to support the arguments.

Authors' response: Thank you for your feedback. Numerous studies have demonstrated that genome editing tools effective in mammalian cells often exhibit significantly reduced editing efficiency or are ineffective in plants, such as xCas9 showed limited activity at non-canonical NGH (H = A, C, T) PAM sites in rice ^{35, 36, 37, 38, 39}, Cas9-NG showed significant reduced activity at the canonical NGG PAM sites in rice ^{38, 39, 40, 41}, LbCas12a in rice cells is higher than the editing efficiency of AsCas12a, which differs from its performance in mammalian cells ^{42, 43, 44}, and SpRY exhibits relatively low editing efficiency in plants in plants ^{7, 45, 46}, and Prime Editor exhibited minimal editing activity initially in plants ^{47, 48, 49}. The referenced studies primarily focused on base editing efficiency in mammalian cells and did not investigate plant cells. In response to this gap, we comprehensively explored the editing performance of multiple editors including TadCBEs and TadDE in plants, conducting research in two crops, rice and tomato, to demonstrate the universality of TadCBEs and TadDE base editors in plants. We hope these references can alleviate your concerns.

You may wonder about what the mechanism behind the difference is observed in different organisms. One major factor that we have observed over the years is temperature. For example, while AsCas12a and LbCas12a showed comparable editing activity in mammalian cells (cultured in 37 °C), AsCas12a showed much lower editing activity than LbCas12a in rice as reported by us ⁴³ and many others as rice is cultured at lower temperature, like 37 °C. Strikingly, LbCas12a nearly failed to show any editing activity in Arabidopsis in our initial test. Hence, in order to achieve genome editing with LbCas12a, one would either use a high-temperature treatment regime as the one developed by our group ⁵⁰ or use a temperature-tolerant LbCas12a (ttLbCas12a) ⁵¹. Recently, we used protein evolution and successfully engineered an improved LbCas12a-

RRV variant for robust genome editing in plants (rice, tomato, and poplar)⁵². Interestingly, we found a different LbCas12a variant, LbCas12a-RVQ, showed optimal activity in human cells⁵². Most of the base editors (including those in this study) are based on Cas9, which is also sensitive to temperature^{50, 53}. Since base editing requires the cooperation of the base excision repair pathway, whose activity may vary among mammals and plants; However, there is currently lacking comprehensive investigation on this. The data from this study and our previous studies (when both protoplasts and stable lines are used for the same constructs for comparison) demonstrate CBEs generally have much higher editing activities than ABEs in plant protoplasts, which are largely non-dividing cells. In stable plants, ABEs especially ABE8e can exert even higher editing efficiency than CBEs. This suggest ABEs mediated A-to-G base conversion largely requires cell replication, while CBEs mediated C-to-T base conversion can rely on both DNA repair (can work in non-dividing cells) and DNA replication. In short, different temperatures and cell types can means quite different genome editing efficiency, which may be observed for the same type of genome editor in different organisms.

3. As noted in the text, other dual base editor CRISPR systems have been reported in plants. These use a nickase Cas9 fused with both a cytidine and adenine deaminase proteins. So such a dual-editing system is not novel in plants. The enzyme is novel (in plants), but as noted above, there are many different types of deaminases reported. The smaller molecular weight is not a convincing argument. The deaminase domains are relatively small (150-230 AAs) compared to cas9 (1,380) and the delivery methods used were not limited by cargo size. There is no way for the smaller molecular weight to be “showcased” in this report.

Authors' response: Thank you for your feedback. It's important to note that previously reported dual base editors require the simultaneous action of two deaminases. For example, the APOBEC3A (199aa) and TadA (166aa) in STEME together constitute 365aa⁵⁴, while the CDA1 (146aa) and TadA9 (166aa) in MoBE together make up 312aa⁵⁵. However, the TadDE dual base editor used in this study consists only of the 166aa TadA-dual deaminase. Compared to previously reported dual base editors, TadDE's

molecular weight is indeed reduced by 146-199 amino acids, a decrease of above 50%. The reduction in molecular weight of the deaminase enzyme is substantial. Considering the complexity of issues such as linkers between multiple components and steric hindrance introduced by multiple structural elements, the advantage of TadDE's lower molecular weight is even more pronounced. Furthermore, the fact that Cpf1 (1200-1300aa) has a smaller protein molecular weight compared to SpCas9 (1368aa) has been widely acknowledged⁴³. TadDE, serving as a dual base editor, indeed exhibits a reduced molecular weight, which can be advantageous for several reasons. This reduction allows TadDE to be combined with more compact Cas proteins, thereby further reducing the size of the base editor and facilitating easier modification of the TadA-dual deaminase to evolve into more effective dual base editors."

In addition, this TadDE has the same or largely overlapping editing window (Fig. 2C and Fig. 3E) for C-to-T editing and A-to-G editing. This is advantageous if one wants to do targeted mutagenesis in a defined window. Also, the editing window is narrow, which explained why we only obtained either C-to-T or A-to-G editing in the two examples that we showed (Fig. 6 and Fig. 7). In these cases, this feature helped minimize by-stander editing outcomes. All these point to the novelty of the demonstration of a new class of dual base editor in plants.

4. Targeting the ALS gene in plants is nothing new. This has been one of the standard genome editing targets since the beginning. SPL14/miR156 is known, but targeting it with base editors is unique as far as I can tell. While gene upregulation was observed, this is a far way away from being a practical application for crop improvement.

Authors' response: Thank you for your inquiry. As far as we understand, many gene editing tools use the OsALS locus as a proof-of-concept application ^{1, 56, 57, 58}, as demonstrated in this study. Though, we did identify some new alleles of OsALS that conferred herbicide resistance. Yes, base editing of OsSPL14, as pointed out by the Review #1, is novel. We have explained its novelty in our opening statement and won't repeat that here. The focus here is to have a proof-of-concept. Also, due to the time

constrain, we did not want to follow up with further analysis the OsSPL14-edited lines for more characterization.

Previous studies on SPL14, involving natural variants, transgenic, and promoter-edited rice plants, have shown that increased SPL14 expression by ~37%-56%⁵⁹, ~60 times⁶⁰, and ~1.5-2.5 times⁶¹ respectively, can result in desirable plant architectures and ultimately lead to higher rice yields. Hence, we have a reason to believe that our SPL14 mutant materials can also contribute to improvements in plant traits, with further investigation. In this study, we identified six SPL14 mutant plants that disrupt the regulation of SPL14 by miRNA156. Specifically, two individual plants retained an unchanged amino acid sequence, and the expression of SPL14 in #B4-16 and #B4-18 increased by 3.5-fold and 2.6-fold, respectively. Based on previous research^{59, 60, 61}, we have a reason to believe that an increase in SPL14 expression alone will result in an ideal plant phenotype. Additionally, numerous studies have shown that miRNA perturbations of target genes can affect plant phenotypes. For example, a 2 bp missense mutation in the coding region of the OsGRF4 transcription factor disrupts the GRF4-miR396 stress response network, leading to increased OsGRF4 transcription levels and resulting in rice plants exhibiting enlarged grains and enhanced cold tolerance⁶². Many studies indicate the applicability of CRISPR-Cas technology for germplasm innovation in crops, emphasizing that materials obtained through gene editing adhere to Mendelian laws of inheritance and can be stably inherited^{63, 64}. Due to similar work about to be published online, which poses competition to our study, we did not present the phenotypes of stably inherited materials due to time constraints. However, the concept is proven here.

Regarding the lethality comment from R1, the authors respond that the combination of low regeneration rates from the multiplex vectors and high editing rates in the single targets indicate lethality due to multiplexing. This is the definition of hand waving. Transformation and regeneration rates are well known in the field to be highly variable. If someone wants to claim anything regarding transformation efficiency, multiple transformation experiments and controls are required. But it appears these claims are based on single experiments without controls. I do not know how one could claim reduced

transformation efficiency based on this experimental setup. Indeed, this experiment was not designed to address this question. I do not understand how the results in supplemental table 3 support the lethality argument. The low frequencies appear to occur across the board, regardless of dual, cytosine or adenine base editor. This observation could very well be due to any number of technical issues. The idea that high rates of base editing leads to lethality is a rather favorable way to interpret these ambiguous results.

Authors' response: Thank you for your thorough review. We agree with the sentiment of your last sentence. The lethality due to high rates of base editing is the favorable hypothesis based on our data. In fact, that points to A-to-G editing based on comparison of C-to-T editing and A-to-G editing in the same T0 lines. This assumption however is based on that our TadDE works well in stable plants. That's why we used single target editing cases to confirm their activity. The fact that we discovered A-to-G editing by all six singular editing constructs in rice stable lines (Fig. 4F) supported this hypothesis (that it is lethality of A-to-G editing of certain target genes among the 10 multiplexed targets skewed the results in multiplexed editing lines).

Our laboratory has many years of experience in stable transformation experiments in rice. In this study, we conducted transformations of both the target construct and the reporter construct simultaneously during rice genetic transformation. Based on the normal transformation efficiency and germination rate observed with the reporter construct, we assessed that the low transformation efficiency and germination rate observed with the multi-gene editing materials, which may include the genes that cause lethality when edited by A-to-G editing. This is a reasonable hypothesis to explain the experimental results. When any surprising data arise, we have to come up with a sensible explanation based on the data. Of course, in the future, one could further investigate by systematically edit these genes one by one and by different combinations. However, we consider this type of work entire fall out of the scope of this current study. After all, any of the findings that we may potentially get will not affect our major conclusion of this study that focuses on tool development. Nevertheless, in the revised manuscript, we have tuned down the statement and emphasized that our hypothesis will need future investigation and cautioned the design of multiplexed genome editing experiments.

We agree with your last two statements. Firstly, we have accumulated experience from previous stable transformation experiments in rice, where we analyzed the stable transformation efficiency and seedling emergence rate across multiple batches and different vectors. The seedling emergence rate in our laboratory consistently ranges from 30% to 50% when using an RFP expression construct (pTX1500) as a transgenic control (Supporting table 1), ruling out low emergence rates due to technical issues. The seedling emergence rate of array 01 edited materials is almost always below 10% (Supplementary table 3, revised), while the emergence rate of TadDE targeting single point mutations remains within the normal range. Thus, we make a reasonable inference about the low transformation efficiency of array 01 vectors. Secondly, the mention of "low transformation efficiency and potential lethality" serves as an analysis of the reasons for choosing array 02 edited materials rather than array 01 edited materials. This explanation does not pertain to the scientific questions and research focus of our manuscript. We demonstrated in rice protoplast experiments that TadCBEa and TadDE exhibit high editing efficiency in both array 01 and array 02. However, in rice plant experiments, due to the low transformation efficiency of array 01 edited plants and insufficient regenerated seedling numbers, we primarily showcase the editing efficiency of array 02. Finally, we would like to emphasize that multi-gene base editing may encounter gene interactions leading to the inability to obtain desired multi-gene edited materials. Actually, such multi-site editing scenarios often lead to lethality or reduced editing efficiency due to genetic interactions, which should be acknowledged and understood. For instance, in our currently study, we found that editing each gene individually within a gene family yielded relatively high editing efficiencies (Supporting figure 2a-b). However, when attempting multi-site editing, the editing efficiency experienced a significant decrease (Supporting figure 2c). In our previous studies, the trends in editing activity between protoplasts and stable transformation were consistent. However, in this study, the trend in editing activity is inconsistent. Referring to previous work, we speculate that the low seedling emergence rate may be due to gene interactions leading to plant lethality. This section has been revised in the manuscript (Supplementary table 3, revised).

Considering that the description in Supplementary Table 3 may not be sufficiently accurate and could potentially be misleading, we have supplemented the original experimental data in the revised supplementary table to better elucidate our results.

Supporting Table 1. Stable transformation efficiency of rice with an RFP expression construct (pTX1500) as a transgenic control.

Experimental batch	Vector	No. of initial callus	resistance callus from first round selection	all regenerated T0 lines	transgenic T0 lines
20230410 batch	pTX1500	30	10; 33.3%	10; 33.3%	10; 100.0%
20230414 batch	pTX1500	35	15; 42.9%	15; 42.9%	14; 93.3%
20230419 batch	pTX1500	35	16; 45.7%	16; 45.7%	15; 93.8%
20230425 batch	pTX1500	35	14; 40.0%	14; 40.0%	14; 100.0%
20230506 batch	pTX1500	35	17; 48.6%	17; 48.6%	17; 100.0%

Supplementary Table 3. The transformation efficiency of multiplexed genome base editing by TadA-8e derived TadCBEs and TadDE in rice T₀ lines. (: revised)**

No.	Vector	Editing type	Target site	No. of initial callus	No. of resistance callus from first round selection	No. of all regenerated T0 lines	No. of transgenic T0 lines
1	pFT261	A3A_Y130F	multiplex array 01	30	15	9	4
2	pFT262	ABE8e	multiplex array 01	60	13	12	1
3	pFT263	TadCBEa	multiplex array 01	60	26	25	8
4	pFT264	TadCBEd	multiplex array 01	60	23	21	5
5	pFT265	TadCBEd_V106W	multiplex array 01	60	55	18	16
6	pFT266	TadDE	multiplex array 01	60	45	2	2
7	pFT272	A3A_Y130F	multiplex array 02	60	50	30	13
8	pFT273	ABE8e	multiplex array 02	60	40	30	9
9	pFT274	TadCBEa	multiplex array 02	60	18	18	14
10	pFT275	TadCBEd	multiplex array 02	60	45	18	14
11	pFT276	TadCBEd_V106W	multiplex array 02	60	45	13	13
12	pFT277	TadDE	multiplex array 02	60	23	18	15

Supporting Figure 2. The possible outcomes of targeted singular editing and targeted multiplexed editing. **A** Editing efficiency of targeted singular editing in rice protoplasts. **B** Plant editing efficiency during targeted singular editing in rice stable transformation. **C** Plant editing efficiency during targeted multiplexed editing in rice stable transformation

I agree with parts of the response regarding the off-target effects. I think R2 is unreasonable to think that off-targets are not relevant for plants. True, one can perform backcrossing. But I agree with the authors that extensive backcrossing is a significant burden for plant breeders. At the same time, the response really oversells what has been reported. Social responsibility?

Authors' response: Thank you for acknowledging our response. To add to this topic, we have provided more background information and elaborated the importance of off-target analysis of genome editors in plants in our opening remarks.

Take together, this research paper thoroughly compares the valuable TadCBEs and TadDE base editor resources recently reported in mammals. It goes beyond those mammalian works by showcasing innovative applications of these tools (e.g. TadDE) in crop engineering and comprehensively assessed off-target effects by using genome-wide and transcriptome-wide sequencing. Such information is critical for researchers to choose and practice genome editing with these new tools.

Reviewer #2 (Remarks to the Author):

This study introduces high-performance cytosine and dual base editors derived from TadA-8e, showing undetectable off-target effects in plants. The research focuses on engineering TadA-8e to achieve both cytosine and dual base editing with high specificity and efficiency in rice and tomato cells. The development of these editors, particularly TadCBEa, TadCBEed, TadCBEed_V106W, and TadDE, is significant for their high editing purity, narrow editing window, and the ability to perform multiplexed base editing in transgenic rice plants without detectable off-target effects through whole genome and transcriptome sequencing. While base editors have demonstrated efficacy in mammalian systems first, their effectiveness in plant systems cannot be assumed solely based on their performance in mammalian systems. Consequently, systematic testing and optimization of base editors in plant systems are crucial for advancing research on plant functional genomics, crop genetic improvement, and the development of new germplasm. The authors' research contributes valuable insights for researchers involved in plant gene editing and crop breeding endeavors.

Authors' response: Thank you for your thorough review, positive feedback, and constructive suggestions. Your time and insights are greatly appreciated.

Reference:

1. Ren Q, *et al.* Improved plant cytosine base editors with high editing activity, purity, and specificity. *Plant Biotechnol J* **19**, 2052-2068 (2021).
2. Randall LB, *et al.* Genome- and transcriptome-wide off-target analyses of an improved cytosine base editor. *Plant Physiol* **187**, 73-87 (2021).
3. Li G, Sretenovic S, Eisenstein E, Coleman G, Qi Y. Highly efficient C - to - T and A - to - G base editing in a Populus hybrid. *Plant Biotechnol J*, (2021).
4. Tang X, *et al.* A large-scale whole-genome sequencing analysis reveals highly specific genome editing by both Cas9 and Cpf1 (Cas12a) nucleases in rice. *Genome Biol* **19**, 84 (2018).
5. Ren Q, *et al.* Improved plant cytosine base editors with high editing activity, purity, and specificity. *Plant Biotechnol J*, (2021).

6. Randall LB, *et al.* Genome- and transcriptome-wide off-target analyses of an improved cytosine base editor. *Plant Physiol*, (2021).
7. Ren Q, *et al.* PAM-less plant genome editing using a CRISPR-SpRY toolbox. *Nat Plants* **7**, 25-33 (2021).
8. Sretenovic S, *et al.* Genome- and transcriptome-wide off-target analyses of a high-efficiency adenine base editor in tomato. *Plant Physiol*, (2023).
9. Wu Y, *et al.* Genome-wide analyses of PAM-relaxed Cas9 genome editors reveal substantial off-target effects by ABE8e in rice. *Plant Biotechnol J* **20**, 1670-1682 (2022).
10. Zhang Y, *et al.* Genome-wide investigation of multiplexed CRISPR-Cas12a-mediated editing in rice. *Plant Genome* **16**, e20266 (2023).
11. Zhang X, *et al.* Increasing the efficiency and targeting range of cytidine base editors through fusion of a single-stranded DNA-binding protein domain. *Nature cell biology* **22**, 740-750 (2020).
12. Tan J, *et al.* PhieABEs: a PAM-less/free high-efficiency adenine base editor toolbox with wide target scope in plants. *Plant Biotechnol J* **20**, 934-943 (2022).
13. Chen L, *et al.* Re-engineering the adenine deaminase TadA-8e for efficient and specific CRISPR-based cytosine base editing. *Nat Biotechnol*, (2022).
14. Neugebauer ME, *et al.* Evolution of an adenine base editor into a small, efficient cytosine base editor with low off-target activity. *Nat Biotechnol*, (2022).
15. Lam DK, *et al.* Improved cytosine base editors generated from TadA variants. *Nat Biotechnol*, (2023).
16. Gaudelli NM, *et al.* Programmable base editing of A*T to G*C in genomic DNA without DNA cleavage. *Nature* **551**, 464-471 (2017).
17. Hua K, Tao X, Yuan F, Wang D, Zhu JK. Precise A.T to G.C Base Editing in the Rice Genome. *Mol Plant* **11**, 627-630 (2018).
18. Li C, *et al.* Expanded base editing in rice and wheat using a Cas9-adenosine deaminase fusion. *Genome Biol* **19**, 59 (2018).
19. Yan F, *et al.* Highly Efficient A.T to G.C Base Editing by Cas9n-Guided tRNA Adenosine Deaminase in Rice. *Mol Plant* **11**, 631-634 (2018).

20. Kleinstiver BP, *et al.* Engineered CRISPR-Cas12a variants with increased activities and improved targeting ranges for gene, epigenetic and base editing. *Nat Biotechnol* **37**, 276-282 (2019).
21. Li X, *et al.* Base editing with a Cpf1-cytidine deaminase fusion. *Nat Biotechnol* **36**, 324-327 (2018).
22. Gaillochot C, *et al.* Systematic optimization of Cas12a base editors in wheat and maize using the ITER platform. *Genome Biol* **24**, 6 (2023).
23. Cheng Y, *et al.* CRISPR-Cas12a base editors confer efficient multiplexed genome editing in rice. *Plant Commun* **4**, 100601 (2023).
24. Zafra MP, *et al.* Optimized base editors enable efficient editing in cells, organoids and mice. *Nat Biotechnol* **36**, 888-893 (2018).
25. Tong H, *et al.* Programmable A-to-Y base editing by fusing an adenine base editor with an N-methylpurine DNA glycosylase. *Nat Biotechnol* **41**, 1080-1084 (2023).
26. Wu X, *et al.* Adenine base editor incorporating the N-methylpurine DNA glycosylase MPGv3 enables efficient A-to-K base editing in rice. *Plant Commun* **4**, 100668 (2023).
27. Li Y, *et al.* Engineering a plant A-to-K base editor with improved performance by fusion with a transactivation module. *Plant Commun* **4**, 100667 (2023).
28. Kurt IC, *et al.* CRISPR C-to-G base editors for inducing targeted DNA transversions in human cells. *Nat Biotechnol* **39**, 41-46 (2021).
29. Zhao D, *et al.* Glycosylase base editors enable C-to-A and C-to-G base changes. *Nat Biotechnol* **39**, 35-40 (2021).
30. Tian Y, *et al.* Efficient C-to-G editing in rice using an optimized base editor. *Plant Biotechnol J* **20**, 1238-1240 (2022).
31. Sretenovic S, *et al.* Exploring C-To-G base editing in rice, tomato, and poplar. *Front Genome Ed* **3**, 756766 (2021).
32. Xu R, *et al.* Genome editing with type II-C CRISPR-Cas9 systems from *Neisseria meningitidis* in rice. *Plant Biotechnol J* **20**, 350-359 (2022).
33. Zong Y, *et al.* Efficient C-to-T base editing in plants using a fusion of nCas9 and human APOBEC3A. *Nat Biotechnol*, (2018).
34. Jin S, *et al.* Rationally designed APOBEC3B cytosine base editors with improved specificity. *Mol Cell* **79**, 728-740 e726 (2020).

35. Hu JH, *et al.* Evolved Cas9 variants with broad PAM compatibility and high DNA specificity. *Nature* **556**, 57-63 (2018).
36. Hua K, Tao X, Han P, Wang R, Zhu JK. Genome Engineering in Rice Using Cas9 Variants that Recognize NG PAM Sequences. *Mol Plant* **12**, 1003-1014 (2019).
37. Wang J, *et al.* xCas9 expands the scope of genome editing with reduced efficiency in rice. *Plant Biotechnol J* **17**, 709-711 (2019).
38. Zhong Z, *et al.* Improving plant genome editing with high-fidelity xCas9 and non-canonical PAM-targeting Cas9-NG. *Mol Plant* **12**, 1027-1036 (2019).
39. Zeng D, *et al.* Engineered Cas9 variant tools expand targeting scope of genome and base editing in rice. *Plant Biotechnol J* **18**, 1348-1350 (2020).
40. Nishimasu H, *et al.* Engineered CRISPR-Cas9 nuclease with expanded targeting space. *Science* **361**, 1259-1262 (2018).
41. Ren B, *et al.* Cas9-NG Greatly Expands the Targeting Scope of the Genome-Editing Toolkit by Recognizing NG and Other Atypical PAMs in Rice. *Mol Plant* **12**, 1015-1026 (2019).
42. Zetsche B, *et al.* Cpf1 is a single RNA-guided endonuclease of a class 2 CRISPR-Cas system. *Cell* **163**, 759-771 (2015).
43. Tang X, *et al.* A CRISPR-Cpf1 system for efficient genome editing and transcriptional repression in plants. *Nat Plants* **3**, 17018 (2017).
44. Tu M, *et al.* A 'new lease of life': FnCpf1 possesses DNA cleavage activity for genome editing in human cells. *Nucleic Acids Res* **45**, 11295-11304 (2017).
45. Walton RT, Christie KA, Whittaker MN, Kleinstiver BP. Unconstrained genome targeting with near-PAMless engineered CRISPR-Cas9 variants. *Science* **368**, 290-296 (2020).
46. Li J, Xu R, Qin R, Liu X, Kong F, Wei P. Genome editing mediated by SpCas9 variants with broad non-canonical PAM compatibility in plants. *Mol Plant* **14**, 352-360 (2021).
47. Anzalone AV, *et al.* Search-and-replace genome editing without double-strand breaks or donor DNA. *Nature* **576**, 149-157 (2019).
48. Lin Q, *et al.* Prime genome editing in rice and wheat. *Nat Biotechnol* **38**, 582-585 (2020).
49. Tang X, *et al.* Plant prime editors enable precise gene editing in rice cells. *Mol Plant* **13**, 667-670 (2020).

50. Malzahn AA, *et al.* Application of CRISPR-Cas12a temperature sensitivity for improved genome editing in rice, maize, and Arabidopsis. *BMC Biol* **17**, 9 (2019).
51. Schindele P, Puchta H. Engineering CRISPR/LbCas12a for highly efficient, temperature-tolerant plant gene editing. *Plant Biotechnol J*, (2019).
52. Zhang L, *et al.* Boosting genome editing efficiency in human cells and plants with novel LbCas12a variants. *Genome Biol* **24**, 102 (2023).
53. LeBlanc C, *et al.* Increased efficiency of targeted mutagenesis by CRISPR/Cas9 in plants using heat stress. *Plant J* **93**, 377-386 (2018).
54. Li C, *et al.* Targeted, random mutagenesis of plant genes with dual cytosine and adenine base editors. *Nat Biotechnol* **38**, 875-882 (2020).
55. Zhang A, *et al.* Directed evolution rice genes with randomly multiplexed sgRNAs assembly of base editors. *Plant Biotechnol J*, (2023).
56. Kuang Y, *et al.* Base-editing-mediated artificial evolution of OsALS1 in planta to develop novel herbicide-tolerant rice germplasms. *Mol Plant* **13**, 565-572 (2020).
57. Zhang R, *et al.* Generating broad-spectrum tolerance to ALS-inhibiting herbicides in rice by base editing. *Sci China Life Sci* **64**, 1624-1633 (2020).
58. Xu R, Kong F, Qin R, Li J, Liu X, Wei P. Development of an efficient plant dual cytosine and adenine editor. *J Integr Plant Biol* **63**, 1600-1605 (2021).
59. Miura K, *et al.* OsSPL14 promotes panicle branching and higher grain productivity in rice. *Nat Genet* **42**, 545-549 (2010).
60. Jiao Y, *et al.* Regulation of OsSPL14 by OsmiR156 defines ideal plant architecture in rice. *Nat Genet* **42**, 541-544 (2010).
61. Song X, *et al.* Targeting a gene regulatory element enhances rice grain yield by decoupling panicle number and size. *Nat Biotechnol* **40**, 1403-1411 (2022).
62. Chen X, *et al.* A missense mutation in Large Grain Size 1 increases grain size and enhances cold tolerance in rice. *J Exp Bot* **70**, 3851-3866 (2019).
63. Wang JY, Doudna JA. CRISPR technology: A decade of genome editing is only the beginning. *Science* **379**, eadd8643 (2023).
64. Gao C. Genome engineering for crop improvement and future agriculture. *Cell* **184**, 1621-1635 (2021).